# Offline Policy Optimization with Variance Regularization

## Abstract

Learning policies from fixed offline datasets is a key challenge to scale up rein-
forcement learning (RL) algorithms towards practical applications. This is often
because off-policy RL algorithms suffer from distributional shift, due to mismatch
between dataset and the target policy, leading to high variance and over-estimation
of value functions. In this work, we propose variance regularization for offline
RL algorithms, using stationary distribution corrections. We show that by using
Fenchel duality, we can avoid double sampling issues for computing the gradient
of the variance regularizer. The proposed algorithm for offline variance regular-
ization (`OVR`) can be used to augment any existing offline policy optimization
algorithms. We show that the regularizer leads to a lower bound to the offline
policy optimization objective, which can help avoid over-estimation errors, and
explains the benefits of our approach across a range of continuous control domains
when compared to existing algorithms.

## 1 Introduction

Offline batch reinforcement learning (RL) algoithms are key towards scaling up RL for real world
applications, such as robotics (Levine et al., 2016) and medical problems . This is because offline
RL provides the appealing ability for agents to learn from fixed datasets, similar to supervised
learning, avoiding continual interaction with the environment, which could be problematic for safety
and feasibility reasons. However, significant mismatch between the fixed collected data and the
policy that the agent is considering can lead to high variance of value function estimates, a problem
encountered by most off-policy RL algorithms (Precup et al., 2000). A complementary problem is
that the value function can become overly optimistic in areas of state space that are outside the visited
batch, leading the agent in data regions where its behavior is poor Fujimoto et al. (2019). Recently
there has been some progress in offline RL (Kumar et al., 2019; Wu et al., 2019b; Fujimoto et al.,
2019), trying to tackle both of these problems.

In this work, we study the problem of offline policy optimization with variance minimization. To
avoid overly optimistic value function estimates, we propose to learn value functions under variance
constraints, leading to a pessimistic estimation, which can significantly help offline RL algorithms,
especially under large distribution mismatch. We propose a framework for variance minimization in
offline RL, such that the obtained estimates can be used to regularize the value function and enable
more stable learning under different off-policy distributions.

We develop a novel approach for variance regularized offline actor-critic algorithms, which we call
`Offline Variance Regularizer` (`OVR`). The key idea of `OVR` is to constrain the policy
improvement step via variance regularized value function estimates. Our algorithmic framework
avoids the double sampling issue that arises when computing gradients of variance estimates, by
instead considering the variance of stationary distribution corrections with per-step rewards, and
using the Fenchel transformation (Boyd & Vandenberghe, 2004) to formulate a minimax optimization
objective. This allows minimizing variance constraints by instead optimizing dual variables, resulting
in simply an augmented reward objective for variance regularized value functions.

We show that even with variance constraints, we can ensure policy improvement guarantees, where the
regularized value function leads to a lower bound on the true value function, which mitigates the usual
overestimation problems in batch RL The use of Fenchel duality in computing the variance allows
us to avoid double sampling, which has been a major bottleneck in scaling up variance-constrained

actor-critic algorithms in prior work A. & Ghavamzadeh (2016); A. & Fu (2018). Practically, our algorithm is easy to implement, since it simply involves augmenting the rewards with the dual variables only, such that the regularized value function can be implemented on top of *any* existing offline policy optimization algorithms. We evaluate our algorithm on existing offline benchmark tasks based on continuous control domains. Our empirical results demonstrate that the proposed variance regularization approach is particularly useful when the batch dataset is gathered at random, or when it is very different from the data distributions encountered during training.

## 2 PRELIMINARIES AND BACKGROUND

We consider an infinite horizon MDP as $(\mathcal{S}, \mathcal{A}, \mathcal{P}, \gamma)$ where $\mathcal{S}$ is the set of states, $\mathcal{A}$ is the set of actions, $\mathcal{P}$ is the transition dynamics and $\gamma$ is the discount factor. The goal of reinforcement learning is to maximize the expected return $\mathcal{J}(\pi) = \mathbb{E}_{s \sim d_\beta}[V^\pi(s)]$, where $V^\pi(s)$ is the value function $V^\pi(s) = \mathbb{E}[\sum_{t=0}^\infty \gamma^t r(s_t, a_t) \mid s_0 = s]$, and $\beta$ is the initial state distribution. Considering parameterized policies $\pi_\theta(a|s)$, the goal is maximize the returns by following the policy gradient (Sutton et al., 1999), based on the performance metric defined as :

$$J(\pi_\theta) = \mathbb{E}_{s_0 \sim \rho, a_0 \sim \pi(s_0)} \Big[ Q^{\pi_\theta}(s_0, a_0) \Big] = \mathbb{E}_{(s,a) \sim d_{\pi_\theta}(s,a)} \Big[ r(s, a) \Big] \tag{1}$$

where $Q^\pi(s, a)$ is the state-action value function, since $V^\pi(s) = \sum_a \pi(a|s) Q^\pi(s, a)$. The policy optimization objective can be equivalently written in terms of the normalized discounted occupancy measure under the current policy $\pi_\theta$, where $d_\pi(s, a)$ is the state-action occupancy measure, such that the normalized state-action visitation distribution under policy $\pi$ is defined as : $d_\pi(s, a) = (1 - \gamma) \sum_{t=0}^\infty \gamma^t P(s_t = s, a_t = a | s_0 \sim \beta, a \sim \pi(s_0))$. The equality in equation 1 holds and can be equivalently written based on the linear programming (LP) formulation in RL (see (Puterman, 1994; Nachum & Dai, 2020) for more details). In this work, we consider the off-policy learning problem under a fixed dataset $\mathcal{D}$ which contains $s, a, r, s'$ tuples under a known behaviour policy $\mu(a|s)$. Under the off-policy setting, importance sampling (Precup et al., 2000) is often used to reweight the trajectory under the behaviour data collecting policy, such as to get unbiased estimates of the expected returns. At each time step, the importance sampling correction $\frac{\pi(a_t|s_t)}{\mu(a_t|s_t)}$ is used to compute the expected return under the entire trajectory as $J(\pi) = (1 - \gamma) \mathbb{E}_{(s,a) \sim d_\mu(s,a)} [\sum_{t=0}^T \gamma^t r(s_t, a_t) \left( \prod_{t=1}^T \frac{\pi(a_t|s_t)}{\mu(a_t|s_t)} \right)]$. Recent works (Fujimoto et al., 2019) have demonstrated that instead of importance sampling corrections, maximizing value functions directly for deterministic or reparameterized policy gradients (Lillicrap et al., 2016; Fujimoto et al., 2018) allows learning under fixed datasets, by addressing the over-estimation problem, by maximizing the objectives of the form $\max_\theta \mathbb{E}_{s \sim \mathcal{D}} \Big[ Q^{\pi_\theta}(s, \pi_\theta(s)) \Big]$.

## 3 VARIANCE REGULARIZATION VIA DUALITY IN OFFLINE POLICY OPTIMIZATION

In this section, we first present our approach based on variance of stationary distribution corrections, compared to importance re-weighting of episodic returns in section 3.1. We then present a derivation of our approach based on Fenchel duality on the variance, to avoid the double sampling issue, leading to a variance regularized offline optimization objective in section 3.2. Finally, we present our algorithm in 1, where the proposed regularizer can be used in any existing offline RL algorithm.

### 3.1 VARIANCE OF REWARDS WITH STATIONARY DISTRIBUTION CORRECTIONS

In this work, we consider the variance of rewards under occupancy measures in offline policy optimization. Let us denote the returns as $D^\pi = \sum_{t=0}^T \gamma^t r(s_t, a_t)$, such that the value function is $V^\pi = \mathbb{E}_\pi[D^\pi]$. The 1-step importance sampling ratio is $\rho_t = \frac{\pi(a_t|s_t)}{\mu(a_t|s_t)}$, and the T-steps ratio can be denoted $\rho_{1:T} = \prod_{t=1}^T \rho_t$. Considering per-decision importance sampling (PDIS) (Precup et al., 2000), the returns can be similarly written as $D^\pi = \sum_{t=0}^T \gamma^t r_t \rho_{0:t}$. The variance of episodic returns, which we denote by $\mathcal{V}_\mathcal{P}(\pi)$, with off-policy importance sampling corrections can be written as :

$$\mathcal{V}_\mathcal{P}(\pi) = \mathbb{E}_{s \sim \beta, a \sim \mu(\cdot|s), s' \sim \mathcal{P}(\cdot|s,a)} \Big[ \Big( D^\pi(s, a) - J(\pi) \Big)^2 \Big].$$

Instead of importance sampling, several recent works have instead proposed for marginalized importance sampling with stationary state-action distribution corrections (Liu et al., 2018; Nachum et al., 2019a; Zhang et al., 2020; Uehara & Jiang, 2019), which can lead to lower variance estimators at the cost of introducing bias. Denoting the stationary distribution ratios as $\omega(s, a) = \frac{d_\pi(s,a)}{d_\mu(s,a)}$, the returns can be written as $W^\pi(s, a) = \omega(s, a) r(s, a)$. The variance of marginalized IS is :

$$\mathcal{V}_\mathcal{D}(\pi) = \mathbb{E}_{(s,a)\sim d_\mu(s,a)}\Big[\Big(W^\pi(s, a) - J(\pi)\Big)^2\Big]$$

$$= \mathbb{E}_{(s,a)\sim d_\mu(s,a)}\Big[W^\pi(s, a)^2\Big] - \mathbb{E}_{(s,a)\sim d_\mu(s,a)}\Big[W^\pi(s, a)\Big]^2 \tag{2}$$

Our key contribution is to first consider the variance of marginalized IS $V_\mathcal{D}(\pi)$ itself a as risk constraints, in the offline batch optimization setting. We show that constraining the offline policy optimization objective with variance of marginalized IS, and using the Fenchel-Legendre transformation on $V_\mathcal{D}(\pi)$ can help avoid the well-known double sampling issue in variance risk constrained RL (for more details on how to compute the gradient of the variance term, see appendix B). We emphasize that the variance here is solely based on returns with occupancy measures, and we do not consider the variance due to the inherent stochasticity of the MDP dynamics.

## 3.2 VARIANCE REGULARIZED OFFLINE MAX-RETURN OBJECTIVE

We consider the variance regularized off-policy max return objective with stationary distribution corrections $\omega_{\pi/\mathcal{D}}$ (which we denote $\omega$ for short for clarity) in the offline fixed dataset $\mathcal{D}$ setting:

$$\max_{\pi_\theta} J(\pi_\theta) := \mathbb{E}_{s\sim\mathcal{D}}\Big[Q^{\pi_\theta}(s, \pi_\theta(s))\Big] - \lambda \mathcal{V}_\mathcal{D}(\omega, \pi_\theta) \tag{3}$$

where $\lambda \geq 0$ allows for the trade-off between offline policy optimization and variance regularization (or equivalently variance risk minimization). The max-return objective under $Q^{\pi_\theta}(s, a)$ has been considered in prior works in offline policy optimization (Fujimoto et al., 2019; Kumar et al., 2019). We show that this form of regularizer encourages variance minimization in offline policy optimization, especially when there is a large data distribution mismatch between the fixed dataset $\mathcal{D}$ and induced data distribution under policy $\pi_\theta$.

## 3.3 VARIANCE REGULARIZATION VIA FENCHEL DUALITY

At first, equation 3 seems to be difficult to optimize, especially for minimizing the variance regularization w.r.t $\theta$. This is because finding the gradient of $\mathcal{V}(\omega, \pi_\theta)$ would lead to the double sampling issue since it contains the squared of the expectation term. The key contribution of OVR is to use the Fenchel duality trick on the second term of the variance expression in equation 2, for regularizing policy optimization objective with variance of marginalized importance sampling. Applying Fenchel duality, $x^2 = \max_y(2xy - y^2)$, to the second term of variance expression, we can transform the variance minimization problem into an equivalent maximization problem, by introducing the dual variables $\nu(s, a)$. We have the Fenchel conjugate of the variance term as :

$$\mathcal{V}(\omega, \pi_\theta) = \max_\nu \quad \Big\{ -\frac{1}{2}\nu(s, a)^2 + \nu(s, a)\omega(s, a)r(s, a) + \mathbb{E}_{(s,a)\sim d_\mathcal{D}}\Big[\omega(s, a)r(s, a)^2\Big]\Big\}$$

$$= \max_\nu \quad \mathbb{E}_{(s,a)\sim d_\mathcal{D}}\Big[ -\frac{1}{2}\nu(s, a)^2 + \nu(s, a)\omega(s, a)r(s, a) + \omega(s, a)r(s, a)^2\Big] \tag{4}$$

Regularizing the policy optimization objective with variance under the Fenchel transformation, we therefore have the overall max-min optimization objective, explicitly written as :

$$\max_\theta \min_\nu J(\pi_\theta, \nu) := \mathbb{E}_{s\sim\mathcal{D}}\Big[Q^{\pi_\theta}(s, \pi_\theta(s))\Big] - \lambda\mathbb{E}_{(s,a)\sim d_\mathcal{D}}\Big[\Big(-\frac{1}{2}\nu^2 + \nu\cdot\omega\cdot r + \omega\cdot r^2\Big)(s, a)\Big] \tag{5}$$

## 3.4 AUGMENTED REWARD OBJECTIVE WITH VARIANCE REGULARIZATION

In this section, we explain the key steps that leads to the policy improvement step being an augmented variance regularized reward objective. The variance minimization step involves estimating the stationary distribution ration (Nachum et al., 2019a), and then simply computing the closed form solution for the dual variables. Fixing dual variables $\nu$, to update $\pi_\theta$, note that this leads to a standard maximum return objective in the dual form, which can be equivalently solved in the primal form,

using augmented rewards. This is because we can write the above above in the dual form as :

$$J(\pi_\theta, \nu, \omega) := \mathbb{E}_{(s,a) \sim d_\mathcal{D}(s,a)} \left[ \omega(s,a) \cdot r(s,a) - \lambda \left( -\frac{1}{2}\nu^2 + \nu \cdot \omega \cdot r + \omega \cdot r^2 \right)(s,a) \right]$$

$$= \mathbb{E}_{(s,a) \sim d_\mathcal{D}(s,a)} \left[ \omega(s,a) \cdot \left( r - \lambda \cdot \nu \cdot r - \lambda \cdot r^2 \right)(s,a) + \frac{\lambda}{2}\nu(s,a)^2 \right]$$

$$= \mathbb{E}_{(s,a) \sim d_\mathcal{D}(s,a)} \left[ \omega(s,a) \cdot \tilde{r}(s,a) + \frac{\lambda}{2}\nu(s,a)^2 \right] \tag{6}$$

where we denote the augmented rewards as :

$$\tilde{r}(s,a) \equiv [r - \lambda \cdot \nu \cdot r - \lambda \cdot r^2](s,a) \tag{7}$$

The policy improvement step can either be achieved by directly solving equation 6 or by considering the primal form of the objective with respect to $Q^{\pi_\theta}(s, \pi_\theta)$ as in (Fujimoto et al., 2019; Kumar et al., 2019). However, solving equation 6 directly can be troublesome, since the policy gradient step involves findinding the gradient w.r.t $\omega(s,a) = \frac{d_{\pi_\theta}(s,a)}{d_\mathcal{D}(s,a)}$ too, where the distribution ratio depends on $d_{\pi_\theta}(s,a)$. This means that the gradient w.r.t $\theta$ would require finding the gradient w.r.t to the normalized discounted occupancy measure, ie, $\nabla_\theta d_{\pi_\theta}(s)$. Instead, it is therefore easier to consider the augmented reward objective, using $\tilde{r}(s,a)$ as in equation 7 in *any* existing offline policy optimization algorithm, where we have the variance regularized value function $\tilde{Q}^{\pi_\theta}(s,a)$.

Note that as highlighted in (Sobel, 1982), the variance of returns follows a Bellman-like equation. Following this, (Bisi et al., 2019) also pointed to a Bellman-like solution for variance w.r.t occupancy measures. Considering variance of the form in equation 2, and the Bellman-like equation for variance, we can write the variance recursively as a Bellman equation:

$$\mathcal{V}^\pi_\mathcal{D}(s,a) = \Big( r(s,a) - J(\pi) \Big)^2 + \gamma \mathbb{E}_{s' \sim \mathcal{P}, a' \sim \pi'(\cdot|s')} \Big[ \mathcal{V}^\pi_\mathcal{D}(s',a') \Big] \tag{8}$$

Since in our objective, we augment the policy improvement step with the variance regularization term, we can write the augmented value function as $Q^\pi_\lambda(s,a) := Q^\pi(s,a) - \lambda \mathcal{V}^\pi_\mathcal{D}(s,a)$. This suggests we can modify existing policy optimization algorithms with augmented rewards on value function.

*Remark :* Applying Fenchel transformation to the variance regularized objective, however, at first glance, seems to make the augmented rewards dependent on the policy itself, since $\tilde{r}(s,a)$ depends on the dual variables $\nu(s,a)$ as well. This can make the rewards non-stationary, thereby the policy maximization step cannot be solved directly via the maximum return objective. However, as we discuss next, the dual variables for minimizing the variance term has a closed form solution $\nu(s,a)$, and thereby does not lead to any non-stationarity in the rewards, due to the alternating minimization and maximization steps.

**Variance Minimization Step :** Fixing the policy $\pi_\theta$, the dual variables $\nu$ can be obtained using closed form solution given by $\nu(s,a) = \omega(s,a) \cdot \tilde{r}(s,a)$. Note that directly optimizing for the target policies using batch data, however, requires a fixed point estimate of the stationary distribution corrections, which can be achieved using existing algorithms (Nachum et al., 2019a; Liu et al., 2018). Solving the optimization objective additionally requires estimating the state-action distribution ratio, $\omega(s,a) = \frac{d_\pi(s,a)}{d_\mathcal{D}(s,a)}$. Recently, several works have proposed estimating the stationary distribution ratio, mostly for the off-policy evaluation case in infinite horizon setting (Zhang et al., 2020; Uehara & Jiang, 2019). We include a detailed discussion of this in appendix E.4.

**Algorithm :** Our proposed variance regularization approach with returns under stationary distribution corrections for offline optimization can be built on top of any existing batch off-policy optimization algorithms. We summarize our contributions in Algorithm 1. Implementing our algorithm requires estimating the state-action distribution ratio, followed by the closed form estimate of the dual variable $\nu$. The augmented stationary reward with the dual variables can then be used to compute the regularized value function $Q^\pi_\lambda(s,a)$. The policy improvement step involves maximizing the variance regularized value function, e.g with BCQ (Fujimoto et al., 2019).

## 4 THEORETICAL ANALYSIS

In this section, we provide theoretical analysis of offline policy optimization algorithms in terms of policy improvement guarantees under fixed dataset $\mathcal{D}$. Following then, we demonstrate that using the variance regularizer leads to a lower bound for our policy optimization objective, which leads to a pessimistic exploitation approach for offline algorithms.

---

**Algorithm 1** `Offline Variance Regularizer`

---

Initialize critic $Q_\phi$, policy $\pi_\theta$, network $\omega_\psi$ and regularization weighting $\lambda$; learning rate $\eta$
**for** $t = 1$ **to** $T$ **do**
    Estimate distribution ratio $\omega_\psi(s, a)$ using any existing DICE algorithm
    Estimate the dual variable $\nu(s, a) = \omega_\psi(s, a) \cdot \tilde{r}(s, a)$
    Calculate augmented rewards $\tilde{r}(s, a)$ using equation 7
    Policy improvement step using *any* offline policy optimization algorithm with augmented
    rewards $\tilde{r}(s, a) : \theta_t = \theta_{t-1} + \eta \nabla_\theta J(\theta, \phi, \psi, \nu)$
**end for**

---

### 4.1 Variance of Marginalized Importance Sampling and Importance Sampling

We first show in lemma 1 that the variance of rewards under stationary distribution corrections can similarly be upper bounded based on the variance of importance sampling corrections. We emphasize that in the off-policy setting under distribution corrections, the variance is due to the estimation of the density ratio compared to the importance sampling corrections.

**Lemma 1.** *The following inequality holds between the variance of per-step rewards under stationary distribution corrections, denoted by $\mathcal{V}_\mathcal{D}(\pi)$ and the variance of episodic returns with importance sampling corrections $\mathcal{V}_\mathcal{P}(\pi)$*

$$\mathcal{V}_\mathcal{P}(\pi) \leq \frac{\mathcal{V}_\mathcal{D}(\pi)}{(1 - \gamma)^2} \tag{9}$$

The proof for this and discussions on the variance of episodic returns compared to per-step rewards under occupancy measures is provided in the appendix B.1.

### 4.2 Policy Improvement Bound under Variance Regularization

In this section, we establish performance improvement guarantees (Kakade & Langford, 2002) for variance regularized value function for policy optimization. Let us first recall that the performance improvement can be written in terms of the total variation $\mathcal{D}_{\text{TV}}$ divergence between state distributions (Touati et al., 2020) (for more discussions on the performance bounds, see appendix C)

**Lemma 2.** *For all policies $\pi'$ and $\pi$, we have the performance improvement bound based on the total variation of the state-action distributions $d_{\pi'}$ and $d_\pi$*

$$J(\pi') \geq \mathcal{L}_\pi(\pi') - \epsilon^\pi \mathcal{D}_{TV}(d_{\pi'} || d_\pi) \tag{10}$$

where $\epsilon^\pi = \max_s |\mathbb{E}_{a \sim \pi'(\cdot|s)}[A^\pi(s, a)]|$, and $\mathcal{L}_\pi(\pi') = J(\pi) + \mathbb{E}_{s \sim d_\pi, a \sim \pi'}[A^\pi(s, a)]$. For detailed proof and discussions, see appendix C. Instead of considering the divergence between state visitation distributions, consider having access to both state-action samples generated from the environment. To avoid importance sampling corrections we can further considers the bound on the objective based on state-action visitation distributions, where we have an upper bound following from (Nguyen et al., 2010) : $D_{\text{TV}}(d_{\pi'}(s) || d_\pi(s)) \leq D_{\text{TV}}(d_{\pi'}(s, a) || d_\pi(s, a))$. Following Pinsker's inequality, we have:

$$J(\pi') \geq J(\pi) + \mathbb{E}_{s \sim d_\pi(s), a \sim \pi'(|s)} \left[ A^\pi(s, a) \right] - \epsilon^\pi \mathbb{E}_{(s,a) \sim d_\pi(s,a)} \left[ \sqrt{\mathcal{D}_{\text{KL}}(d_{\pi'}(s, a) || d_\pi(s, a))} \right] \tag{11}$$

Furthermore, we can exploit the relation between KL, total variation (TV) and variance through the variational representation of divergence measures. Recall that the total divergence between P and Q distributions is given by : $\mathcal{D}_{\text{TV}}(p, q) = \frac{1}{2} \sum_x |p(x) - q(x)|$. We can use the variational representation of the divergence measure. Denoting $d_\pi(s, a) = \beta_{\pi'}(s, a)$, we have

$$D_{\text{TV}}(\beta_{\pi'} || \beta_\pi) = \sup_{f : \mathcal{S} \times \mathcal{A} \rightarrow \mathbb{R}} \left[ \mathbb{E}_{(s,a) \sim \beta_{\pi'}}[f(s, a)] - \mathbb{E}_{(s,a) \sim \beta(s,a)}[\phi^* \circ f(s, a)] \right] \tag{12}$$

where $\phi^*$ is the convex conjugate of $\phi$ and $f$ is the dual function class based on the variational representation of the divergence. Similar relations with the variational representations of f-divergences have also been considered in (Nachum et al., 2019b; Touati et al., 2020). We can finally obtain a bound for the policy improvement following this relation, in terms of the per-step variance:

**Theorem 1.** *For all policies $\pi$ and $\pi'$, and the corresponding state-action visitation distributions $d_{\pi'}$ and $d_\pi$, we can obtain the performance improvement bound in terms of the variance of rewards under state-action occupancy measures.*

$$J(\pi') - J(\pi) \geq \mathbb{E}_{s \sim d_\pi(s), a \sim \pi'(a|s)}[A^\pi(s, a)] - Var_{(s,a) \sim d_\pi(s,a)} \left[ f(s, a) \right] \tag{13}$$

*where $f(s, a)$ is the dual function class from the variational representation of variance.*

*Proof.* For detailed proof, see appendix C.1. □

### 4.3 LOWER BOUND OBJECTIVE WITH VARIANCE REGULARIZATION

In this section, we show that augmenting the policy optimization objective with a variance regularizer leads to a lower bound to the original optimization objectiven $J(\pi_\theta)$. Following from (Metelli et al., 2018), we first note that the variance of marginalized importance weighting with distribution corrections can be written in terms of the $\alpha-$Renyi divergence. Let p and q be two probability measures, such that the Renyi divergence is $\mathcal{F}_\alpha = \frac{1}{\alpha} \log \sum_x q(x) \left(\frac{p(x)}{q(x)}\right)^\alpha$. When $\alpha = 1$, this leads to the well-known KL divergence $\mathcal{F}_1(p||q) = \mathcal{F}_{\text{KL}}(p||q)$.

Let us denote the state-action occupancy measures under $\pi$ and dataset $\mathcal{D}$ as $d_\pi$ and $d_\mathcal{D}$. The variance of state-action distribution ratios is $\text{Var}_{(s,a) \sim d_\mathcal{D}(s,a)}[\omega_{\pi/\mathcal{D}}(s, a)]$. When $\alpha = 2$ for the Renyi divergence, we have :

$$\text{Var}_{(s,a) \sim d_\mathcal{D}(s,a)}[\omega_{\pi/\mathcal{D}}(s, a)] = \mathcal{F}_2(d_\pi||d_\mathcal{D}) - 1 \tag{14}$$

Following from (Metelli et al., 2018), and extending results from importance sampling $\rho$ to marginalized importance sampling $\omega_{\pi/\mathcal{D}}$, we provide the following result that bounds the variance of the approximated density ratio $\hat{\omega}_{\pi/\mathcal{D}}$ in terms of the Renyi divergence :

**Lemma 3.** *Assuming that the rewards of the MDP are bounded by a finite constant, $||r||_\infty \leq R_{max}$. Given random variable samples $(s, a) \sim d_\mathcal{D}(s, a)$ from dataset $\mathcal{D}$, for any $N > 0$, the variance of marginalized importance weighting can be upper bounded as :*

$$Var_{(s,a) \sim d_\mathcal{D}(s,a)}\left[\hat{\omega}_{\pi/\mathcal{D}}(s, a)\right] \leq \frac{1}{N}||r||_\infty^2 \mathcal{F}_2(d_\pi||d_\mathcal{D}) \tag{15}$$

See appendix D.1 for more details. Following this, our goal is to derive a lower bound objective to our off-policy optimization problem. Concentration inequalities has previously been studied for both off-policy evaluation (Thomas et al., 2015a) and optimization (Thomas et al., 2015b). In our case, we can adapt the concentration bound derived from Cantelli's ineqaulity and derive the following result based on variance of marginalized importance sampling. Under state-action distribution corrections, we have the following lower bound to the off-policy policy optimization objective with stationary state-action distribution corrections

**Theorem 2.** *Given state-action occupancy measures $d_\pi$ and $d_\mathcal{D}$, and assuming bounded reward functions, for any $0 < \delta \leq 1$ and $N > 0$, we have with probability at least $1 - \delta$ that :*

$$J(\pi) \geq \mathbb{E}_{(s,a) \sim d_\mathcal{D}(s,a)}\left[\omega_{\pi/\mathcal{D}}(s, a) \cdot r(s, a)\right] - \sqrt{\frac{1 - \delta}{\delta} Var_{(s,a) \sim d_\mathcal{D}(s,a)}[\omega_{\pi/\mathcal{D}}(s, a) \cdot r(s, a)]} \tag{16}$$

Equation 16 shows the lower bound policy optimization objective under risk-sensitive variance constraints. The key to our derivation in equation 16 of theorem 2 shows that given off-policy batch data collected with behaviour policy $\mu(a|s)$, we are indeed optimizing a lower bound to the policy optimization objective, which is regularized with a variance term to minimize the variance in batch off-policy learning.

## 5 EXPERIMENTAL RESULTS ON BENCHMARK OFFLINE CONTROL TASKS

*Experimental Setup :* We demonstrate the significance of variance regularizer on a range of continuous control domains (Todorov et al., 2012) based on fixed offline datasets from (Fu et al., 2020), which is a standard benchmark for offline algorithms. To demonstrate the significance of our variance regularizer OVR, we mainly use it on top of the BCQ algorithm and compare it with other existing baselines, using the benchmark D4RL (Fu et al., 2020) offline datasets for different tasks and off-policy distributions. Experimental results are given in table 1

**Performance on Optimal and Medium Quality Datasets :** We first evaluate the performance of OVR when the dataset consists of optimal and mediocre logging policy data. We collected the dataset using a fully (*expert*) or partially (*medium*) trained SAC policy. We build our algorithm OVR on top of BCQ, denoted by BCQ + VAR. Note that the OVR algorithm can be agnostic to the behaviour policy too for computing the distribution ratio (Nachum et al., 2019a) and the variance. We observe that even

| Domain | Task Name | BCQ+OVR | BCQ | BEAR | BRAC-p | aDICE | SAC-off |
|---|---|---|---|---|---|---|---|
| Gym | halfcheetah-random | 0.00 | 0.00 | 25.1 | 24.1 | -0.3 | **30.5** |
| | hopper-random | 9.51 | 9.65 | **11.4** | 11 | 0.9 | 11.3 |
| | walker-random | 5.16 | 0.48 | **7.3** | -0.2 | 0.5 | 4.1 |
| | halfcheetah-medium | 35.6 | 34.9 | 41.7 | **43.8** | -2.2 | -4.3 |
| | hopper-medium | **71.24** | 57.76 | 52.1 | 32.7 | 1.2 | 0.9 |
| | walker-medium | 33.90 | 27.13 | 59.1 | **77.5** | 0.3 | 0.8 |
| | halfcheetah-expert | **100.02** | 97.99 | - | - | - | - |
| | hopper-expert | **108.41** | 98.36 | - | - | - | - |
| | walker-expert | 71.77 | **72.93** | - | - | - | - |
| | halfcheetah-medium-expert | **59.52** | 54.12 | 53.4 | 44.2 | -0.8 | 1.8 |
| | hopper-medium-expert | 44.68 | 37.20 | **96.3** | 1.9 | 1.1 | -0.1 |
| | walker-medium-expert | 34.53 | 29.00 | 40.1 | **76.9** | 0.4 | 1.6 |
| | halfcheetah-mixed | **29.95** | 29.91 | - | - | - | - |
| | hopper-mixed | **16.36** | 10.88 | - | - | - | - |
| | walker-mixed | **14.74** | 10.23 | - | - | - | - |
| FrankaKitchen | kitchen-complete | 4.48 | 3.38 | 0 | 0 | 0 | **15** |
| | kitchen-partial | **25.65** | 19.11 | 13.1 | 0 | 0 | 0 |
| | kitchen-mixed | 30.59 | 23.55 | **47.2** | 0 | 2.5 | 2.5 |

Table 1: The results on D4RL tasks compare BCQ (Fujimoto et al., 2019) with and without OVR, bootstrapping error reduction (BEAR) (Kumar et al., 2019), behavior-regularized actor critic with policy (BRAC-p) (Wu et al., 2019a), AlgeaDICE (aDICE) (Nachum et al., 2019b) and offline SAC (SAC-off) (Haarnoja et al., 2018). The results presented are the normalized returns on the task as per Fu et al. (2020) (see Table 3 in Fu et al. (2020) for the unnormalized scores on each task). We see that in most tasks we are able to significant gains using OVR. Our algorithm can be applied to any policy optimization baseline algorithm that trains the policy by maximizing the expected rewards. Unlike BCQ, BEAR (Kumar et al., 2019) does not have the same objective, as they train the policy using and MMD objective.

though performance is marginally improved with OVR under expert settings, since the demonstrations are optimal itself, we can achieve significant improvements under medium dataset regime. This is because OVR plays a more important role when there is larger variance due to distribution mismatch between the data logging and target policy distributions. Experimental results are shown in first two columns of figure 1.

**Performance on Random and Mixed Datasets :** We then evaluate the performance on *random* datasets, i.e, the worst-case setup when the data logging policy is a random policy, as shown in the last two columns of figure 1. As expected, we observe no improvements at all, and even existing baselines such as BCQ (Fujimoto et al., 2019) can work poorly under random dataset setting. When we collect data using a mixture of random and mediocre policy, denoted by *mixed*, the performance is again improved for OVR on top of BCQ, especially for the Hopper and Walker control domains. We provide additional experimental results and ablation studies in appendix E.1.

## 6 RELATED WORKS

We now discuss related works in offline RL, for evaluation and opimization, and its relations to variance and risk sensitive algorithms. We include more discussions of related works in appendix A.1. In off-policy evaluation, per-step importance sampling (Precup et al., 2000; 2001) have previously been used for off-policy evaluation function estimators. However, this leads to high variance estimators, and recent works proposed using marginalized importance sampling, for estimating stationary state-action distribution ratios (Liu et al., 2018; Nachum et al., 2019a; Zhang et al., 2019), to reduce variance but with additional bias. In this work, we build on the variance of marginalized IS, to develop variance risk sensitive offline policy optimization algorithm. This is in contrast to prior works on variance constrained online actor-critic (A. & Ghavamzadeh, 2016; Chow et al., 2017; Castro et al., 2012) and relates to constrained policy optimization methods (Achiam et al., 2017; Tessler et al., 2019).

For offline policy optimization, several works have recently addressed the overestimation problem in batch RL (Fujimoto et al., 2019; Kumar et al., 2019; Wu et al., 2019b), including the very recently proposed Conservative Q-Learning (CQL) algorithm (Kumar et al., 2020). Our work is done in parallel to CQL, due to which we do not include it as a baseline in our experiments. CQL learns a value function which is guaranteed to lower-bound the true value function. This helps prevent value over-estimation for out-of-distribution (OOD) actions, which is an important issue in offline RL. We

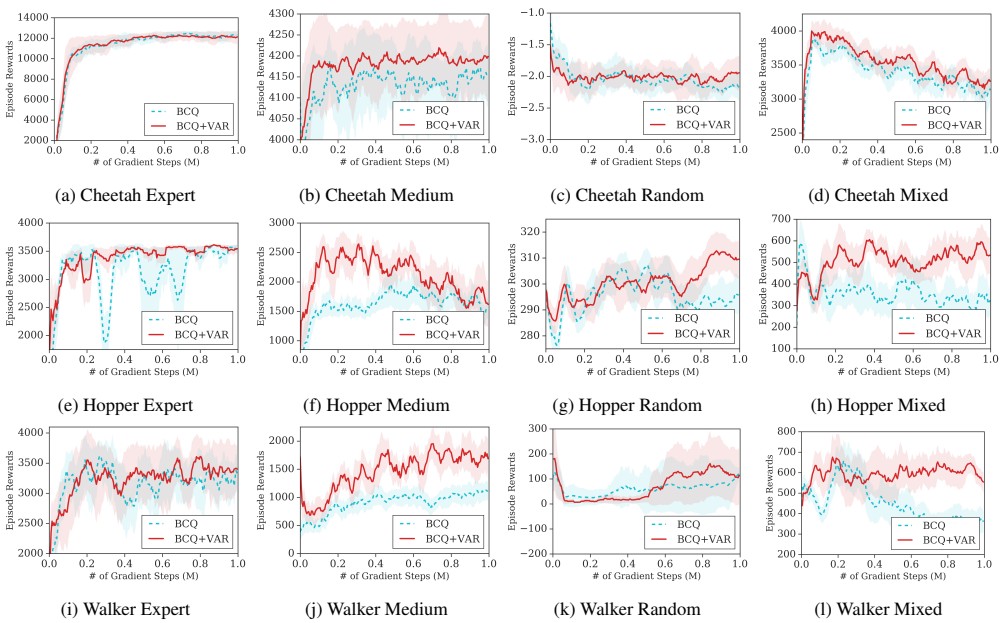

Figure 1: Evaluation of the proposed approach and the baseline BCQ (Fujimoto et al., 2019) on a suite of three OpenAI Gym environments. Details about the type of offline dataset used for training, namely *random*, *medium*, *mixed*, and *expert* are included in Appendix. Results are averaged over 5 random seeds (Henderson et al., 2018). We evaluate the agent using standard procedures, as in Kumar et al. (2019); Fujimoto et al. (2019)
.

note that our approach is orthogonal to CQL in that CQL introduces a regularizer on the state action value function $Q^\pi(s, a)$ based on the Bellman error (the first two terms in equation 2 of CQL), while we introduce a variance regularizer on the stationary state distribution $d_\pi(s)$. Since the value of a policy can be expressed in two ways - either through $Q^\pi(s, a)$ or occupancy measures $d_\pi(s)$, both CQL and our paper are essentially motivated by the same objective of optimizing a lower bound on $J(\theta)$, but through different regularizers. Our work can also be considered similar to AlgaeDICE (Nachum et al., 2019b), since we introduce a variance regularizer based on the distribution corrections, instead of minimizing the f-divergence between stationary distributions in AlgaeDICE. Both our work and AlgaeDICE considers the dual form of the policy optimization objective in the batch setting, where similar to the Fenchel duality trick on our variance term, AlgaeDICE instead uses the variational form, followed by the change of variables tricks, inspired from (Nachum et al., 2019a) to handle their divergence measure.

## 7  DISCUSSION AND CONCLUSION

We proposed a new framework for offline policy optimization with variance regularization called OVR, to tackle high variance issues due to distribution mismatch in offline policy optimization. Our work provides a practically feasible variance constrained actor-critic algorithm that avoids double sampling issues in prior variance risk sensitive algorithms (Castro et al., 2012; A. & Ghavamzadeh, 2016). The presented variance regularizer leads to a lower bound to the true offline optimization objective, thus leading to pessimistic value function estimates, avoiding both high variance and overestimation problems in offline RL. Experimentally, we evaluate the significance of OVR on standard benchmark offline datasets, with different data logging off-policy distributions, and show that OVR plays a more significant role when there is large variance due to distribution mismatch. While we only provide a variance related risk sensitive approach for offline RL, for future work, it would be interesting other risk sensitive approaches (Chow & Ghavamzadeh, 2014; Chow et al., 2017) and examine its significance in batch RL. We hope our proposed variance regularization framework would provide new opportunities for developing practically robust risk sensitive offline algorithms.

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

## A   APPENDIX : ADDITIONAL DISCUSSIONS

### A.1   EXTENDED RELATED WORK

**Other related works :**  Several other prior works have previously considered the batch RL setting (Lange et al., 2012) for off-policy evaluation, counterfactual risk minimization (Swaminathan & Joachims, 2015a;b), learning value based methods such as DQN (Agarwal et al., 2019), and others (Kumar et al., 2019; Wu et al., 2019b). Recently, batch off-policy optimization has also been introduced to reduce the exploitation error (Fujimoto et al., 2019) and for regularizing with arbitrary behaviour policies (Wu et al., 2019b). However, due to the per-step importance sampling corrections on episodic returns (Precup et al., 2000), off-policy batch RL methods is challenging. In this work, we instead consider marginalized importance sampling corrections and correct for the stationary state-action distributions (Nachum et al., 2019a; Uehara & Jiang, 2019; Zhang et al., 2020). Additionally, under the framework of Constrained MDPs (Altman & Asingleutility, 1999), risk-sensitive and constrained actor-critic algorithms have been proposed previously (Chow et al., 2017; Chow & Ghavamzadeh, 2014; Achiam et al., 2017). However, these works come with their own demerits, as they mostly require minimizing the risk (ie, variance) term, where finding the gradient of the variance term often leads a double sampling issue (Baird, 1995). We avoid this by instead using Fenchel duality (Boyd & Vandenberghe, 2004), inspired from recent works (Nachum & Dai, 2020; Dai et al., 2018) and cast risk constrained actor-critic as a max-min optimization problem. Our work is closely related to (Bisi et al., 2019), which also consider per-step variance of returns, w.r.t state occupancy measures in the on-policy setting, while we instead consider the batch off-policy optimization setting with per-step rewards w.r.t stationary distribution corrections.

Constrained optimization has previously been studied in in reinforcement learning for batch policy learning (Le et al., 2019), and optimization (Achiam et al., 2017), mostly under the framework of constrained MDPs (Altman & Asingleutility, 1999). In such frameworks, the cumulative return objective is augmented with a set of constraints, for safe exploration (García et al., 2015; Perkins & Barto, 2003; Ding et al., 2020), or to reduce risk measures (Chow et al., 2017; A. & Fu, 2018; Castro et al., 2012). Batch learning algorithms (Lange et al., 2012) have been considered previously for counterfactual risk minimization and generalization (Swaminathan & Joachims, 2015a;b) and policy evaluation (Thomas et al., 2015a; Li et al., 2015), although little has been done for constrained offline policy based optimization. This raises the question of how can we learn policies in RL from fixed offline data, similar to supervised or unsupervised learning.

### A.2   WHAT MAKES OFFLINE OFF-POLICY OPTIMIZATION DIFFICULT?

Offline RL optimization algorithms often suffer from distribution mismatch issues, since the underlying data distribution in the batch data may be quite different from the induced distribution under target policies. Recent works (Fujimoto et al., 2019; Kumar et al., 2019; Agarwal et al., 2019; Kumar et al., 2020) have tried to address this, by avoiding overestimation of Q-values, which leads to the extraplation error when bootstrapping value function estimates. This leads to offline RL agents generalizing poorly for unseen regions of the dataset. Additionally, due to the distribution mismatch, value function estimates can also have large variance, due to which existing online off-policy algorithms (Haarnoja et al., 2018; Lillicrap et al., 2016; Fujimoto et al., 2018) may fail without online interactions with the environment. In this work, we address the later problem to minimize variance of value function estimates through variance related risk constraints.

## B   APPENDIX : PER-STEP VERSUS EPISODIC VARIANCE OF RETURNS

Following from (Castro et al., 2012; A. & Ghavamzadeh, 2016), let us denote the returns with importance sampling corrections in the off-policy learning setting as :

$$D^\pi(s,a) = \sum_{t=0}^{T} \gamma^t r(s_t, a_t) \Big( \prod_{t=1}^{T} \frac{\pi(a_t \mid s_t)}{\mu(a_t \mid s_t)} \Big) \mid s_0 = s, a_0 = a, \tau \sim \mu \qquad (17)$$

From this definition in equation 17, the action-value function, with off-policy trajectory-wise importance correction is $Q^\pi(s,a) = \mathbb{E}_{(s,a) \sim d_\mu(s,a)}[D^\pi(s,a)]$, and similarly the value function can be defined as : $V^\pi(s) = \mathbb{E}_{s \sim d_\mu(s)}[D^\pi(s)]$. For the trajectory-wise importance corrections, we can

define the variance of the returns, similar to (A. & Fu, 2018) as :

$$\mathcal{V}_{\mathcal{P}}(\pi) = \mathbb{E}_{(s,a)\sim d_\mu(s,a)}[D^\pi(s,a)^2] - \mathbb{E}_{(s,a)\sim d_\mu(s,a)}[D^\pi(s,a)]^2 \tag{18}$$

where note that as in (Sobel, 1982), equation 18 also follows a Bellman like equation, although due to lack of monotonocitiy as required for dynamic programming (DP), such measures cannot be directly optimized by standard DP algorithms (A. & Fu, 2018).

In contrast, if we consider the variance of returns with stationary distribution corrections (Nachum et al., 2019a; Liu et al., 2018), rather than the product of importance sampling ratios, the variance term involves weighting the rewards with the distribution ratio $\omega_{\pi/\mu}$. Typically, the distribution ratio is approximated using a separate function class (Uehara & Jiang, 2019), such that the variance can be written as :

$$W^\pi(s,a) = \omega_{\pi/\mathcal{D}}(s,a) \cdot r(s,a) \mid s = s, a \sim \pi(\cdot \mid s), (s,a) \sim d_{\mathcal{D}}(s,a) \tag{19}$$

where we denote $\mathcal{D}$ as the data distribution in the fixed dataset, collected by either a known or unknown behaviour policy. The variance of returns under occupancy measures is therefore given by :

$$\mathcal{V}_{\mathcal{D}}(\pi) = \mathbb{E}_{(s,a)\sim d_{\mathcal{D}}(s,a)}\Big[W^\pi(s,a)^2\Big] - \mathbb{E}_{(s,a)\sim d_{\mathcal{D}}(s,a)}\Big[W^\pi(s,a)\Big]^2 \tag{20}$$

where note that the variance expression in equation 20 depends on the square of the per-step rewards with distribution correction ratios. We denote this as the dual form of the variance of returns, in contrast to the primal form of the variance of expected returns (Sobel, 1982).

Note that even though the variance term under episodic per-step importance sampling corrections in equation 18 is equivalent to the variance with stationary distribution corrections in equation 20, following from (Bisi et al., 2019), considering per-step corrections, we will show that the variance with distribution corrections indeed upper bounds the variance of importance sampling corrections. This is an important relationship, since constraining the policy improvement step under variance constraints with occupancy measures therefore allows us to obtain a lower bound to the offline optimization objective, similar to (Kumar et al., 2020).

## B.1 PROOF OF LEMMA 1 : VARIANCE INEQUALITY

Following from (Bisi et al., 2019), we show that the variance of per-step rewards under occupancy measures, denoted by $\mathcal{V}_{\mathcal{D}}(\pi)$ upper bounds the variance of episodic returns $\mathcal{V}_{\mathcal{P}}(\pi)$.

$$\mathcal{V}_{\mathcal{P}}(\pi) \leq \frac{\mathcal{V}_{\mathcal{D}}(\pi)}{(1-\gamma)^2} \tag{21}$$

*Proof.* Proof of Lemma 1 following from (Bisi et al., 2019) is as follows. Denoting the returns, as above, but for the on-policy case with trajectories under $\pi$, as $D^\pi(s,a) = \sum_{t=0}^\infty \gamma^t r(s_t, a_t)$, and denoting the return objective as $J(\pi) = \mathbb{E}_{s_0\sim\rho, a_t\sim\pi(\cdot|s_t), s'\sim\mathcal{P}}\Big[D^\pi(s,a)\Big]$, the variance of episodic returns can be written as :

$$\mathcal{V}_{\mathcal{P}}(\pi) = \mathbb{E}_{(s,a)\sim d_\pi(s,a)}\Big[\Big(D^\pi(s,a) - \frac{J(\pi)}{(1-\gamma)}\Big)^2\Big] \tag{22}$$

$$= \mathbb{E}_{(s,a)\sim d_\pi(s,a)}\Big[(D^\pi(s,a))^2\Big] + \frac{J(\pi)}{(1-\gamma)^2} - \frac{2J(\pi)}{(1-\gamma)}\mathbb{E}_{(s,a)\sim d_\pi(s,a)}\Big[D^\pi(s,a)\Big] \tag{23}$$

$$= \mathbb{E}_{(s,a)\sim d_\pi(s,a)}\Big[D^\pi(s,a)^2\Big] - \frac{J(\pi)^2}{(1-\gamma)^2} \tag{24}$$

Similarly, denoting returns under occupancy measures as $W^\pi(s,a) = d_\pi(s,a)r(s,a)$, and the returns under occupancy measures, equivalently written as $J(\pi) = \mathbb{E}_{(s,a)\sim d_\pi(s,a)}[r(s,a)]$ based on the primal and dual forms of the objective (Uehara & Jiang, 2019; Nachum & Dai, 2020), we can equivalently write the variance as :

$$\mathcal{V}_{\mathcal{D}}(\pi) = \mathbb{E}_{(s,a)\sim d_\pi(s,a)}\Big[\Big(r(s,a) - J(\pi)\Big)^2\Big] \tag{25}$$

$$= \mathbb{E}_{(s,a)\sim d_\pi(s,a)}\Big[r(s,a)^2\Big] + J(\pi)^2 - 2J(\pi)\mathbb{E}_{(s,a)\sim d_\pi(s,a)}[r(s,a)] \tag{26}$$

$$= \mathbb{E}_{(s,a)\sim d_\pi(s,a)}\Big[r(s,a)^2\Big] - J(\pi)^2 \tag{27}$$

Following from equation 22 and 25, we therefore have the following inequality :

$$(1-\gamma)^2 \mathbb{E}_{s_0 \sim \rho, a \sim \pi}\Big[D^\pi(s,a)^2\Big] \le (1-\gamma)^2 \mathbb{E}_{s_0 \sim \rho, a \sim \pi}\Big[\Big(\sum_{t=0}^\infty \gamma^t\Big)\Big(\sum_{t=0}^\infty \gamma^t r(s_t,a_t)^2\Big)\Big] \quad (28)$$

$$= (1-\gamma)\mathbb{E}_{s_0 \sim \rho, a \sim \pi}\Big[\sum_{t=0}^\infty \gamma^t r(s_t,a_t)^2\Big] \quad (29)$$

$$= \mathbb{E}_{(s,a)\sim d_\pi(s,a)}\Big[r(s,a)^2\Big] \quad (30)$$

where the first line follows from Cauchy-Schwarz inequality. This concludes the proof. □

We can further extend lemma 1, for off-policy returns under stationary distribution corrections (ie, marginalized importance sampling) compared importance sampling. Recall that we denote the variance under stationary distribution corrections as :

$$\mathcal{V}_\mathcal{D}(\pi) = \mathbb{E}_{(s,a)\sim d_\mathcal{D}(s,a)}\Big[\Big(\omega_{\pi/\mathcal{D}}(s,a)\cdot r(s,a) - J(\pi)\Big)^2\Big] \quad (31)$$

$$= \mathbb{E}_{(s,a)\sim d_\mathcal{D}(s,a)}\Big[\omega_{\pi/\mathcal{D}}(s,a)^2 \cdot r(s,a)^2\Big] - J(\pi)^2 \quad (32)$$

where $J(\pi) = \mathbb{E}_{(s,a)\sim d_\mathcal{D}(s,a)}\Big[\omega_{\pi/\mathcal{D}}(s,a)\cdot r(s,a)\Big]$. We denote the episodic returns with importance sampling corrections as : $D^\pi = \sum_{t=0}^T \gamma^t r_t \rho_{0:t}$. The variance, as denoted earlier is given by :

$$\mathcal{V}_\mathcal{P}(\pi) = \mathbb{E}_{(s,a)\sim d_\pi(s,a)}\Big[D^\pi(s,a)^2\Big] - \frac{J(\pi)^2}{(1-\gamma)^2} \quad (33)$$

We therefore have the following inequality

$$(1-\gamma)^2 \mathbb{E}_{s_0 \sim \rho, a \sim \pi}\Big[D^\pi(s,a)^2\Big] \le (1-\gamma)^2 \mathbb{E}_{s_0 \sim \rho, a \sim \pi}\Big[\Big(\sum_{t=0}^T \gamma^t\Big)\Big(\sum_{t=0}^T \gamma^t r(s_t,a_t)^2\Big)\Big(\prod_{t=0}^T \frac{\pi(a_t|s_t)}{\mu_\mathcal{D}(a_t|s_t)}\Big)^2\Big]$$

$$= (1-\gamma)\mathbb{E}_{s_0 \sim \rho, a \sim \pi}\Big[\sum_{t=0}^\infty \gamma^t r(s_t,a_t)^2 \Big(\prod_{t=0}^T \frac{\pi(a_t|s_t)}{\mu_\mathcal{D}(a_t|s_t)}\Big)^2\Big] \quad (34)$$

$$= \mathbb{E}_{(s,a)\sim d_\mathcal{D}(s,a)}\Big[\omega_{\pi/\mathcal{D}}(s,a)^2 \cdot r(s,a)^2\Big] \quad (35)$$

which shows that lemma 1 also holds for off-policy returns with stationary distribution corrections.

## B.2 DOUBLE SAMPLING FOR COMPUTING GRADIENTS OF VARIANCE

The gradient of the variance term often leads to the double sampling issue, thereby making it impractical to use. This issue has also been pointed out by several other works (A. & Ghavamzadeh, 2016; Castro et al., 2012; Chow et al., 2017), since the variance involves the squared of the objective function itself. Recall that we have:

$$\mathcal{V}_\mathcal{D}(\theta) = \mathbb{E}_{(s,a)\sim d_\mathcal{D}}\Big[\omega_{\pi/\mathcal{D}}(s,a)\cdot r(s,a)^2\Big] - \Big\{\mathbb{E}_{(s,a)\sim d_\mathcal{D}}\Big[\omega_{\pi/\mathcal{D}}(s,a)\cdot r(s,a)\Big]\Big\}^2 \quad (36)$$

The gradient of the variance term is therefore :

$$\nabla_\theta \mathcal{V}_\mathcal{D}(\theta) = \nabla_\theta \mathbb{E}_{(s,a)\sim d_\mathcal{D}}\Big[\omega_{\pi/\mathcal{D}}(s,a)\cdot r(s,a)^2\Big]$$

$$- 2\cdot\Big\{\mathbb{E}_{(s,a)\sim d_\mathcal{D}}\Big[\omega_{\pi/\mathcal{D}}(s,a)\cdot r(s,a)\Big]\Big\}\cdot \nabla_\theta\Big\{\mathbb{E}_{(s,a)\sim d_\mathcal{D}}\Big[\omega_{\pi/\mathcal{D}}(s,a)\cdot r(s,a)\Big]\Big\} \quad (37)$$

where equation 37 requires multiple samples to compute the expectations in the second term. To see why this is true, let us denote

$$J(\theta) = \mathbb{E}_{d_\mathcal{D}(s,a)}\Big[\underbrace{\omega_{\pi/\mathcal{D}}(s,a)}\cdot r(s,a)_{\text{IS}(\omega,\pi_\theta)}\Big]$$

where we have $\text{IS}(\omega, \pi_\theta)$ as the returns in short form. The variance of the returns with the stationary state-action distribution corrections can therefore be written as :

$$\mathcal{V}_\mathcal{D}(\theta) = \mathbb{E}_{d_\mathcal{D}(s,a)}\underbrace{\Big[\text{IS}(\omega,\pi_\theta)^2\Big]}_{(a)} - \underbrace{\mathbb{E}_{d_\mathcal{D}(s,a)}\Big[\text{IS}(\omega,\pi_\theta)\Big]^2}_{(b)} \quad (38)$$

We derive the gradient of each of the terms in (a) and (b) in equation 38 below. First, we find the gradient of the variance term w.r.t $\theta$ :

$$
\begin{aligned}
\nabla_\theta \mathbb{E}_{d_\mathcal{D}(s,a)}\Big[\mathrm{IS}(\omega, \pi_\theta)^2\Big] &= \nabla_\theta \sum_{s,a} d_\mathcal{D}(s,a)\mathrm{IS}(\omega, \pi_\theta)^2 = \sum_{s,a} d_\mathcal{D}(s,a)\nabla_\theta \mathrm{IS}(\omega, \pi_\theta)^2 \\
&= \sum_{s,a} d_\mathcal{D}(s,a) \cdot 2 \cdot \mathrm{IS}(\omega, \pi_\theta) \cdot \mathrm{IS}(\omega, \pi_\theta) \cdot \nabla_\theta \log \pi_\theta(a \mid s) \\
&= 2 \cdot \sum_{s,a} d_\mathcal{D}(s,a)\mathrm{IS}(\omega, \pi_\theta)^2 \nabla_\theta \log \pi_\theta(a \mid s) \\
&= 2 \cdot \mathbb{E}_{d_\mathcal{D}(s,a)}\Big[\mathrm{IS}(\omega, \pi_\theta)^2 \cdot \nabla_\theta \log \pi_\theta(a \mid s)\Big]
\end{aligned}
\tag{39}
$$

Equation 39 interestingly shows that the variance of the returns w.r.t $\pi_\theta$ has a form similar to the policy gradient term, except the critic estimate in this case is given by the importance corrected returns, since $\mathrm{IS}(\omega, \pi_\theta) = [\omega_{\pi/\mathcal{D}}(s,a) \cdot r(s,a)]$. We further find the gradient of term (b) from equation 38. Finding the gradient of this second term w.r.t $\theta$ is therefore :

$$
\nabla_\theta \mathbb{E}_{d_\mathcal{D}(s,a)}\Big[\mathrm{IS}(\omega, \pi_\theta)\Big]^2 = \nabla_\theta J(\theta)^2 = 2 \cdot J(\theta) \cdot \mathbb{E}_{d_\mathcal{D}(s,a)}\Big[\omega_{\pi/\mathcal{D}} \cdot \{\nabla_\theta \log \pi_\theta(a \mid s) \cdot Q^\pi(s,a)\}\Big]
\tag{40}
$$

Overall, the expression for the gradient of the variance term is therefore :

$$
\begin{aligned}
\nabla_\theta \mathcal{V}_\mathcal{D}(\theta) = 2 \cdot \mathbb{E}_{d_\mathcal{D}(s,a)}\Big[\mathrm{IS}(\omega, \pi_\theta)^2 \cdot \nabla_\theta \log \pi_\theta(a \mid s)\Big] \\
- 2 \cdot J(\theta) \cdot \mathbb{E}_{d_\mathcal{D}(s,a)}\Big[\omega_{\pi/\mathcal{D}} \cdot \{\nabla_\theta \log \pi_\theta(a \mid s) \cdot Q^\pi(s,a)\}\Big]
\end{aligned}
\tag{41}
$$

The variance gradient in equation 41 is difficult to estimate in practice, since it involves both the gradient of the objective and the objective $J(\theta)$ itself. This is known to have the double sampling issue (Baird, 1995) which requires separate independent rollouts. Previously, (Castro et al., 2012) tackled the variance of the gradient term using simultaneous perturbation stochastic approximation (SPSA) (Spall, 1992), where we can keep running estimates of both the return and the variance term, and use a two time scale algorithm for computing the gradient of the variance regularizer with per-step importance sampling corrections.

## B.3 Alternative Derivation : Variance Regularization via Fenchel Duality

In the derivation of our algorithm, we applied the Fenchel duality trick to the second term of the variance expression 25. An alternative way to derive the proposed algorithm would be to see what happens if we apply the Fenchel duality trick to both terms of the variance expression. This might be useful since equation 41 requires evaluating both the gradient terms and the actual objective $J(\theta)$, due to the analytical expression of the form $\nabla_\theta J(\theta) \cdot J(\theta)$, hence suffering from a double sampling issue. In general, the Fenchel duality is given by :

$$
x^2 = \max_y(2xy - y^2)
\tag{42}
$$

and applying Fenchel duality to both the terms, since they both involve squared terms, we get :

$$
\begin{aligned}
\mathbb{E}_{d_\mathcal{D}(s,a)}\Big[\mathrm{IS}(\omega, \pi_\theta)^2\Big] &\equiv \mathbb{E}_{d_\mathcal{D}(s,a)}\Big[\max_y \Big\{2 \cdot \mathrm{IS}(\omega, \pi_\theta) \cdot y(s,a) - y(s,a)^2\Big\}\Big] \\
&= 2 \cdot \max_y \Big\{\mathbb{E}_{d_\mathcal{D}(s,a)}\Big[\mathrm{IS}(\omega, \pi_\theta) \cdot y(s,a)\Big] - \mathbb{E}_{d_\mathcal{D}(s,a)}\Big[y(s,a)^2\Big]\Big\}
\end{aligned}
\tag{43}
$$

Similarly, applying Fenchel duality to the second (b) term we have :

$$
\mathbb{E}_{d_\mathcal{D}(s,a)}\Big[\mathrm{IS}(\omega, \pi_\theta)\Big]^2 = \max_\nu \Big\{2 \cdot \mathbb{E}_{d_\mathcal{D}(s,a)}\Big[\mathrm{IS}(\omega, \pi_\theta) \cdot \nu(s,a)\Big] - \nu^2\Big\}
\tag{44}
$$

Overall, we therefore have the variance term, after applying Fenchel duality as follows, leading to an overall objective in the form $\max_y \max_\nu \mathcal{V}_\mathcal{D}(\theta)$, which we can use as our variance regularizer

$$
\begin{aligned}
\mathcal{V}_\mathcal{D}(\theta) = 2 \cdot \max_y \Big\{\mathbb{E}_{d_\mathcal{D}(s,a)}\Big[\mathrm{IS}(\omega, \pi_\theta) \cdot y(s,a)\Big] - \mathbb{E}_{d_\mathcal{D}(s,a)}\Big[y(s,a)^2\Big]\Big\} \\
- \max_\nu \Big\{2 \cdot \mathbb{E}_{d_\mathcal{D}(s,a)}\Big[\mathrm{IS}(\omega, \pi_\theta) \cdot \nu(s,a)\Big] - \nu^2\Big\}
\end{aligned}
\tag{45}
$$

Using the variance of stationary distribution correction returns as a regularizer, we can find the gradient of the variance term w.r.t $\theta$ as follows, where the gradient terms dependent on the dual variables $y$ and $\nu$ are 0.

$$\nabla_\theta \mathcal{V}_\mathcal{D}(\theta) = 2 \cdot \nabla_\theta \mathbb{E}_{d_\mathcal{D}(s,a)}\Big[\text{IS}(\omega, \pi_\theta) \cdot y(s,a)\Big] - 0 - 2 \cdot \nabla_\theta \mathbb{E}_{d_\mathcal{D}(s,a)}\Big[\text{IS}(\omega, \pi_\theta) \cdot \nu(s,a)\Big] + 0$$

$$= 2 \cdot \mathbb{E}_{d_\mathcal{D}(s,a)}\Big[\text{IS}(\omega, \pi_\theta) \cdot y(s,a) \cdot \nabla_\theta \log \pi_\theta(a \mid s)\Big] - 2 \cdot \mathbb{E}_{d_\mathcal{D}(s,a)}\Big[\text{IS}(\omega, \pi_\theta) \cdot \nu(s,a) \cdot \nabla_\theta \log \pi_\theta(a \mid s)\Big]$$

$$= 2 \cdot \mathbb{E}_{d_\mathcal{D}(s,a)}\Big[\text{IS}(\omega, \pi_\theta) \cdot \nabla_\theta \log \pi_\theta(a \mid s) \cdot \Big\{y(s,a) - \nu(s,a)\Big\}\Big] \quad (46)$$

Note that from equation 46, the two terms in the gradient is almost equivalent, and the difference comes only from the difference between the two dual variables $y(s,a)$ and $\nu(s,a)$. Note that our variance term also requires separately maximizing the dual variables, both of which has the following closed form updates :

$$\nabla_\nu \mathcal{V}_\mathcal{D}(\theta) = -2 \cdot \nabla_\nu \mathbb{E}_{d_\mathcal{D}(s,a)}\Big[\text{IS}(\omega, \pi_\theta) \cdot \nu(s,a)\Big] + \nabla_\nu \nu^2 = 0 \quad (47)$$

Solving which exactly, leads to the closed form solution $\nu(s,a) = \mathbb{E}_{d_\mathcal{D}(s,a)}\Big[\text{IS}(\omega, \pi_\theta)\Big]$. Similarly, we can also solve exactly using a closed form solution for the dual variables $y$, such that :

$$\nabla_y \mathcal{V}_\mathcal{D}(\theta) = 2 \cdot \nabla_y \mathbb{E}_{d_\mathcal{D}(s,a)}\Big[\text{IS}(\omega, \pi_\theta) \cdot y(s,a)\Big] - 2 \cdot \nabla_y \mathbb{E}_{d_\mathcal{D}(s,a)}\Big[y(s,a)^2\Big] = 0 \quad (48)$$

Solving which exactly also leads to the closed form solution, such that $y(s,a) = \frac{1}{2} \cdot \text{IS}(\omega, \pi_\theta) = \frac{1}{2} \cdot \frac{d_\pi(s,a)}{d_\mu(s,a)} \cdot r(s,a)$. Note that the exact solutions for the two dual variables are similar to each other, where $\nu(s,a)$ is the expectation of the returns with stationary distribution corrections, whereas $y(s,a)$ is only the return from a single rollout.

## C  Appendix : Monotonic Performance Improvement Guarantees under Variance Regularization

We provide theoretical analysis and performance improvements bounds for our proposed variance constrained policy optimization approach. Following from (Kakade & Langford, 2002; Schulman et al., 2015; Achiam et al., 2017), we extend existing performance improvement guarantees based on the stationary state-action distributions instead of only considering the divergence between the current policy and old policy. We show that existing conservative updates in algorithms (Schulman et al., 2015) can be considered for both state visitation distributions and the action distributions, as similarly pointed by (Achiam et al., 2017). We can then adapt this for the variance constraints instead of the divergence constraints. According to the performance difference lemma (Kakade & Langford, 2002), we have that, for all policies $\pi$ and $\pi'$ :

$$J(\pi') - J(\pi) = \mathbb{E}_{s \sim d_{\pi'}, a \sim \pi'}[A^\pi(s,a)] \quad (49)$$

which implies that when we maximize 49, it will lead to an improved policy $\pi'$ with policy improvement guarantees over the previous policy $\pi$. We can write the advantage function with variance augmented value functions as :

$$A_\lambda^\pi = Q_\lambda^\pi(s,a) - V_\lambda^\pi(s) = \mathbb{E}_{s' \sim \mathcal{P}}\Big[r(s,a) - \lambda(r(s,a) - J(\pi))^2 + \gamma V_\lambda^\pi(s') - V_\lambda^\pi(s)\Big]$$

However, equation 49 is often difficult to maximize directly, since it additionally requires samples from $\pi'$ and $d_{\pi'}$, and often a surrogate objective is instead proposed by (Kakade & Langford, 2002). Following (Schulman et al., 2015), we can therefore obtain a bound for the performance difference based on the variance regularized advantage function :

$$J(\pi') \geq J(\pi) + \mathbb{E}_{s \sim d_\pi(s), a \sim \pi'(a|s)}\Big[A_\lambda^\pi(s,a)\Big] \quad (50)$$

where we have the augmented rewards for the advantage function, and by following Fenchel duality for the variance, can avoid policy dependent reward functions. Otherwise, we have the augmented rewards for value functions as $\tilde{r}(s,a) = r(s,a) - \lambda(r(s,a) - J(\pi))^2$. This however suggests that the performance difference does not hold without proper assumptions (Bisi et al., 2019). We can therefore obtain a monotonic improvement guarantee by considering the KL divergence between

policies :

$$\mathcal{L}_\pi(\pi') = J(\pi) + \mathbb{E}_{s\sim d_\pi, a\sim\pi'}[A^\pi(s,a)] \tag{51}$$

which ignores the changes in the state distribution $d_{\pi'}$ due to the improved policy $\pi'$. (Schulman et al., 2015) optimizes the surrogate objectives $\mathcal{L}_\pi(\pi')$ while ensuring that the new policy $\pi'$ stays close to the current policy $\pi$, by imposing a KL constraint $(\mathbb{E}_{s\sim d_\pi}[\mathcal{D}_{KL}(\pi'(\cdot \mid s)||\pi(\cdot \mid s)] \leq \delta)$. The performance difference bound, based on the constraint between $\pi$ and $\pi'$ as in TRPO (Schulman et al., 2015) is given by :

**Lemma 4.** *The performance difference lemma in (Schulman et al., 2015), where* $\alpha = \mathcal{D}_{TV}^{max} = \max_s \mathcal{D}_{TV}(\pi, \pi')$

$$J(\pi') \geq \mathcal{L}_\pi(\pi') - \frac{4\epsilon\gamma}{(1-\gamma)^2}(\mathcal{D}_{TV}^{max}(\pi'||\pi))^2 \tag{52}$$

*where* $\epsilon = \max_{s,a}|A^\pi(s,a)|$, *which is usually denoted with* $\alpha$, *where*

The performance improvement bxound in (Schulman et al., 2015) can further be written in terms of the KL divergence by following the relationship between total divergence (TV) and KL, which follows from Pinsker's inequality, $\mathcal{D}_{TV}(p||q)^2 \leq \mathcal{D}_{KL}(p||q)$, to get the following improvement bound :

$$J(\pi') \geq \mathcal{L}_\pi(\pi') - \frac{4\epsilon\gamma}{(1-\gamma)^2}\mathcal{D}_{KL}(\pi'||\pi) \tag{53}$$

We have a performance difference bound in terms of the state distribution shift $d_{\pi'}$ and $d_\pi$. This justifies that $\mathcal{L}_\pi(\pi')$ is a sensible lower bound to $J(\pi')$ as long as there is a total variation distance between $d_{\pi'}$ and $d_\pi$ which ensures that the policies $\pi'$ and $\pi$ stay close to each other. Finally, following from (Achiam et al., 2017), we obtain the following lower bound, which satisfies policy improvement guarantees :

$$J(\pi') \geq \mathcal{L}_\pi(\pi') - \frac{2\gamma\epsilon^\pi}{1-\gamma}\mathbb{E}_{s\sim d_\pi}[\mathcal{D}_{TV}(\pi'(\cdot \mid s)||\pi(\cdot \mid s))] \tag{54}$$

Equation 53 and 54 assumes that there is no state distribution shift between $\pi'$ and $\pi$. However, if we explicitly assume state distribution changes, $d_{\pi'}$ and $d_\pi$ due to $\pi'$ and $\pi$ respectively, then we have the following performance improvement bound :

**Lemma 5.** *For all policies* $\pi'$ *and* $\pi$, *we have the performance improvement bound based on the total variation of the state-action distributions* $d_{\pi'}$ *and* $d_\pi$

$$J(\pi') \geq \mathcal{L}_\pi(\pi') - \epsilon^\pi \mathcal{D}_{TV}(d_{\pi'}||d_\pi) \tag{55}$$

*where* $\epsilon^\pi = \max_s |\mathbb{E}_{a\sim\pi'(\cdot|s)}[A^\pi(s,a)]|$

which can be further written in terms of the surrogate objective $\mathcal{L}_\pi(\pi')$ as :

$$J(\pi') \geq J(\pi) + \mathbb{E}_{s\sim d_\pi, a\sim\pi'}[A^\pi(s,a)] - \epsilon^\pi\mathcal{D}_{TV}(d_{\pi'}||d_\pi)$$
$$= \mathcal{L}_\pi(\pi') - \epsilon^\pi\mathcal{D}_{TV}(d_{\pi'}||d_\pi) \tag{56}$$

### C.1 PROOF OF THEOREM 1 : POLICY IMPROVEMENT BOUND WITH VARIANCE REGULARIZATION

*Proof.* We provide derivation for theorem 1. Recall that for all policies $\pi'$ and $\pi$, and corresponding state visitation distributions $d_{\pi'}$ and $d_\pi$, we can obtain the performance improvement bound in terms of the variance of state-action distribution corrections

$$J(\pi') - J(\pi) \geq \mathbb{E}_{s\sim d_\pi, a\sim\pi'}\left[A^\pi(s,a)\right] - \text{Var}_{s\sim d_\pi, a\sim\pi}\left[f(s,a)\right] \tag{57}$$

where $f(s,a)$ is the dual function class, for the divergence between $d_{\pi'}(s,a)$ and $d_\pi(s,a)$ Following from Pinsker's inequality, the performance difference lemma written in terms of the state visitation distributions can be given by :

$$J(\pi') \geq \mathcal{L}_\pi(\pi') - \epsilon^\pi\mathcal{D}_{TV}(d_{\pi'}||d_\pi)$$
$$\geq J(\pi) + \mathbb{E}_{s\sim d_\pi, a\sim\pi'}[A^\pi(s,a)] - \epsilon^\pi\mathcal{D}_{TV}(d_{\pi'}||d_\pi)$$
$$\geq J(\pi) + \mathbb{E}_{s\sim d_\pi, a\sim\pi'}[A^\pi(s,a)] - \epsilon^\pi\sqrt{\mathcal{D}_{KL}(d_{\pi'}||d_\pi)} \tag{58}$$

Following from (Schulman et al., 2015), we can alternately write this follows, where we further apply the variational form of TV

$$J(\pi') \geq J(\pi) + \mathbb{E}_{s \sim d_\pi, a \sim \pi'}\Big[A^\pi(s,a)\Big] - C \cdot \mathbb{E}_{s \sim d_\pi}\Big[\mathcal{D}_{\mathrm{TV}}(d_{\pi'}||d_\pi)^2\Big]$$

$$= J(\pi) + \mathbb{E}_{s \sim d_\pi, a \sim \pi'}\Big[A^\pi(s,a)\Big] - C \cdot \mathbb{E}_{s \sim d_\pi}\Big[\Big(\max_f\{\mathbb{E}_{s \sim d_{\pi'}, a \sim \pi}[f(s,a)] - \mathbb{E}_{s \sim d_\pi, a \sim \pi}[f(s,a)]\}\Big)^2\Big]$$

$$\geq J(\pi) + \mathbb{E}_{s \sim d_\pi, a \sim \pi'}\Big[A^\pi(s,a)\Big] - C \cdot \max_f \mathbb{E}_{s \sim d_\pi}\Big[\Big(\mathbb{E}_{s \sim d_{\pi'}, a \sim \pi}[f(s,a)] - \mathbb{E}_{s \sim d_\pi, a \sim \pi}[f(s,a)]\Big)^2\Big]$$

$$= J(\pi) + \mathbb{E}_{s \sim d_\pi, a \sim \pi'}\Big[A^\pi(s,a)\Big] - C \cdot \max_f \Big\{\Big(\mathbb{E}_{s \sim d_\pi, a \sim \pi}[f(s,a)] - \mathbb{E}_{s \sim d_\pi, a \sim \pi}[\mathbb{E}_{s \sim d_\pi, a \sim \pi}[f(s,a)]]\Big)^2\Big\}$$

$$= J(\pi) + \mathbb{E}_{s \sim d_\pi, a \sim \pi'}\Big[A^\pi(s,a)\Big] - C \cdot \max_f \mathrm{Var}_{s \sim d_\pi, a \sim \pi}\Big[f(s,a)\Big] \tag{59}$$

Therefore the policy improvement bound depends on maximizing the variational representation $f(s,a)$ of the f-divergence to guaranetee improvements from $J(\pi)$ to $J(\pi')$. This therefore leads to the stated result in theorem 1. $\qquad\square$

# D    APPENDIX : LOWER BOUND OBJECTIVE WITH VARIANCE REGULARIZATION

## D.1    PROOF OF LEMMA 3

Recalling lemma 3 which states that, the proof of this follows from (Metelli et al., 2018). We extend this for marginalized importance weighting, and include here for completeness. Note that compared to importance weighting, which leads to an unbiased estimator as in (Metelli et al., 2018), correcting for the state-action occupancy measures leads to a biased estimator, due to the approximation $\hat{\omega}_{\pi/\mathcal{D}}$. However, for our analysis, we only require to show a lower bound objective, and therefore do not provide any bias variance analysis as in off-policy evaluation.

$$\mathrm{Var}_{(s,a) \sim d_\mathcal{D}(s,a)}\Big[\hat{\omega}_{\pi/\mathcal{D}}\Big] \leq \frac{1}{N}||r||_\infty^2 \mathcal{F}_2(d_\pi||d_\mathcal{D}) \tag{60}$$

*Proof.* Assuming that state action samples are drawn i.i.d from the dataset $\mathcal{D}$, we can write :

$$\mathrm{Var}_{(s,a) \sim d_\mathcal{D}(s,a)}\Big[\hat{\omega}_{\pi/\mathcal{D}}(s,a)\Big] \leq \frac{1}{N}\mathrm{Var}_{(s_1,a_1) \sim d_\mathcal{D}(s,a)}\Big[\frac{d_\pi(s_1,a_1)}{d_\mathcal{D}(s_1,a_1)} \cdot r(s_1,a_1)\Big]$$

$$\leq \frac{1}{N}\mathbb{E}_{(s_1,a_1) \sim d_\mathcal{D}(s,a)}\Big[\Big(\frac{d_\pi(s_1,a_1)}{d_\mathcal{D}(s_1,a_1)} \cdot r(s_1,a_1)\Big)^2\Big]$$

$$\leq \frac{1}{N}||r||_\infty^2 \mathbb{E}_{(s_1,a_1) \sim d_\mathcal{D}(s,a)}\Big[\Big(\frac{d_\pi(s_1,a_1)}{d_\mathcal{D}(s_1,a_1)} \cdot r(s_1,a_1)\Big)^2\Big] = \frac{1}{N}||r||_\infty^2 \mathcal{F}_2(d_\pi||d_\mathcal{D}) \tag{61}$$

$\qquad\square$

## D.2    PROOF OF THEOREM 2:

First let us recall the stated theorem 2. By constraining the off-policy optimization problem with variance constraints, we have the following lower bound to the optimization objective with stationary state-action distribution corrections

$$J(\pi) \geq \mathbb{E}_{(s,a) \sim d_\mathcal{D}(s,a)}[\frac{d_\pi(s,a)}{d_\mathcal{D}(s,a)}r(s,a)] - \sqrt{\frac{1-\delta}{\delta}\mathrm{Var}_{(s,a) \sim d_\mu(s,a)}[\frac{d_\pi(s,a)}{d_\mathcal{D}(s,a)}r(s,a)]} \tag{62}$$

*Proof.* The proof for the lower bound objective can be obtained as follows. We first define a relationship between the variance and the $\alpha$-divergence with $\alpha = 2$, as also similarly noted in (Metelli et al., 2018). Given we have batch samples $\mathcal{D}$, and denoting the state-action distribution correction with $\omega_{\pi/\mathcal{D}}(s,a)$, we can write from lemma 3 :

$$\mathrm{Var}_{(s,a) \sim d_\mathcal{D}(s,a)}\Big[\hat{\omega}_{\pi/\mathcal{D}}\Big] \leq \frac{1}{N}||r||_\infty^2 \mathcal{F}_2(d_\pi||d_\mathcal{D}) \tag{63}$$

where the per-step estimator with state-action distribution corrections is given by $\omega_{\pi/\mathcal{D}}(s,a) \cdot r(s,a)$. Here, the reward function $r(s,a)$ is a bounded function, and for any $N > 0$ the variance of the

per-step reward estimator with distribution corrections can be upper bounded by the Renyi-divergence ($\alpha = 2$). Finally, following from (Metelli et al., 2018) and using Cantelli's inequality, we have with probability at least $1 - \delta$ where $0 < \delta < 1$ :

$$\Pr\Big(\omega_{\pi/\mathcal{D}} - J(\pi) \geq \lambda\Big) \leq \frac{1}{1 + \frac{\lambda^2}{\text{Var}_{(s,a)\sim d_{\mathcal{D}}(s,a)}[\omega_{\pi/\mathcal{D}}(s,a)\cdot r(s,a)]}} \tag{64}$$

and by using $\delta = \frac{1}{1 + \frac{\lambda^2}{\text{Var}_{(s,a)\sim d_{\mathcal{D}}(s,a)}[\omega_{\pi/\mathcal{D}}(s,a)\cdot r(s,a)]}}$ we get that with probability at least $1 - \delta$, we have:

$$J(\pi) = \mathbb{E}_{(s,a)\sim d_{\pi}(s,a)} \geq \mathbb{E}_{(s,a)\sim d_{\mathcal{D}}(s,a)}[\omega_{\pi/\mathcal{D}}(s,a) \cdot r(s,a)] - \sqrt{\frac{1-\delta}{\delta}\text{Var}_{(s,a)\sim d_{\mathcal{D}}(s,a)}[\omega_{\pi/\mathcal{D}}(s,a) \cdot r(s,a)]} \tag{65}$$

where we can further replace the variance term with $\alpha = 2$ for the Renyi divergence to conclude the proof for the above theorem. We can further write the lower bound for for $\alpha$-Renyi divergence, following the relation between variance and Renyi-divergence for $\alpha = 2$ as :

$$J(\pi) = \mathbb{E}_{(s,a)\sim d_{\pi}(s,a)}[r(s,a)] \geq \mathbb{E}_{(s,a)\sim d_{\mathcal{D}}(s,a)}[\frac{d_{\pi}(s,a)}{d_{\mathcal{D}}(s,a)} \cdot r(s,a)] - ||r||_{\infty}\sqrt{\frac{(1-\delta)d_2(d_{\pi}||d_{\mathcal{D}})}{\delta N}}$$

This hints at the similarity between our proposed variance regularized objective with that of other related works including AlgaeDICE (Nachum et al., 2019b) which uses a f-divergence $D_{f(d_{\pi}||d_{\mathcal{D}})}$ between stationary distributions. □

# E  APPENDIX : ADDITIONAL EXPERIMENTAL RESULTS

## E.1  EXPERIMENTAL ABLATION STUDIES

In this section, we present additional results using state-action experience replay weightings on existing offline algorithms, and analysing the significance of our variance regularizer on likelihood corrected offline algorithms. Denoting $\omega(s, a)$ for the importance weighting of state-action occupancy measures based on samples in the experience replay buffer, we can modify existing offline algorithms to account for state-action distribution ratios.

The ablation experimental results using the Hopper control benchmark are summarized in figure 2. The same base BCQ algorithm is used with a modified objective for BCQ (Fujimoto et al., 2019) where the results for applying off-policy importance weights are denoted as "BCQ+I.W.". We employ the same technique to obtain $\omega(s, a)$ for both the baseline and for adding variance regularization as described. The results suggest that adding the proposed per-step variance regularization scheme significantly outperforms just importance weighting the expected rewards for off-policy policy learning.

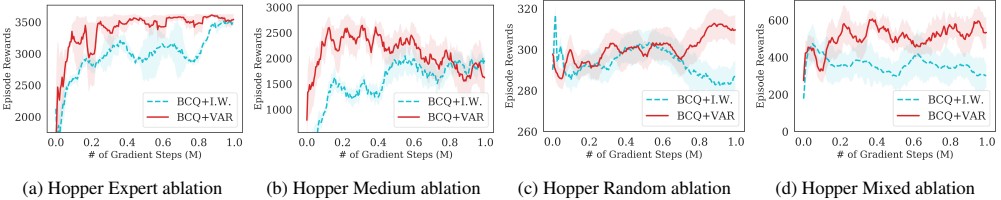

(a) Hopper Expert ablation     (b) Hopper Medium ablation     (c) Hopper Random ablation     (d) Hopper Mixed ablation

Figure 2: Ablation performed on Hopper. The mean and standard deviation are reported over 5 random seeds. The offline datasets for these experiments are same as the corresponding ones in Fig 1 of the main paper.

## E.2  EXPERIMENTAL RESULTS IN CORRUPTED NOISE SETTINGS

We additionally consider a setting where the batch data is collected from a noisy environment, i.e, in settings with *corrupted rewards*, $r \rightarrow r + \epsilon$, where $\epsilon \sim \mathcal{N}(0, 1)$. Experimental results are presented in figures 1, 3. From our results, we note that using OVR on top of BCQ (Fujimoto et al., 2019), we can achieve significantly better performance with variance minimization, especially when the agent is given sub-optimal demonstrations. We denote it as *medium* (when the dataset was collected by a half trained SAC policy) or a *mixed* behaviour logging setting (when the data logging policy is a mixture of random and SAC policy). This is also useful for practical scalability, since often data collection is

| Domain | Task Name | BCQ+OVR | BCQ | BEAR | BRAC-p | aDICE | SAC-off |
|--------|-----------|---------|-----|------|--------|-------|---------|
| Adroit | pen-human | **64.12** | 56.58 | -1 | 8.1 | -3.3 | 6.3 |
| | hammer-human | **1.05** | 0.75 | 0.3 | 0.3 | 0.3 | 0.5 |
| | door-human | 0.00 | 0.00 | -0.3 | -0.3 | 0 | **3.9** |
| | relocate-human | -0.13 | **-0.08** | -0.3 | -0.3 | -0.1 | 0 |
| | pen-cloned | 40.84 | **41.09** | 26.5 | 1.6 | -2.9 | 23.5 |
| | hammer-cloned | **0.78** | 0.35 | 0.3 | 0.3 | 0.3 | 0.2 |
| | door-cloned | **0.03** | **0.03** | -0.1 | -0.1 | 0 | 0 |
| | relocate-cloned | -0.22 | -0.26 | -0.3 | -0.3 | -0.3 | **-0.2** |
| | pen-expert | 99.32 | 89.42 | **105.9** | -3.5 | -3.5 | 6.1 |
| | hammer-expert | **119.32** | 108.38 | 127.3 | 0.3 | 0.3 | 25.2 |
| | door-expert | **100.39** | 101.33 | **103.4** | -0.3 | 0 | 7.5 |
| | relocate-expert | 31.31 | 23.55 | **98.6** | -0.3 | -0.1 | -0.3 |

Table 2: The results on D4RL tasks compare BCQ (Fujimoto et al., 2019) with and without OVR, bootstrapping error reduction (BEAR) (Kumar et al., 2019), behavior-regularized actor critic with policy (BRAC-p) (**?**), AlgeaDICE (aDICE) (Nachum et al., 2019b) and offline SAC (SAC-off) (Haarnoja et al., 2018). The results presented are the normalized returns on the task as per Fu et al. (2020) (see Table 3 in Fu et al. (2020) for the unnormalized scores on each task).

expensive from an expert policy. We add noise to the dataset, to examine the significance of OVR under a noisy corrupted dataset setting.

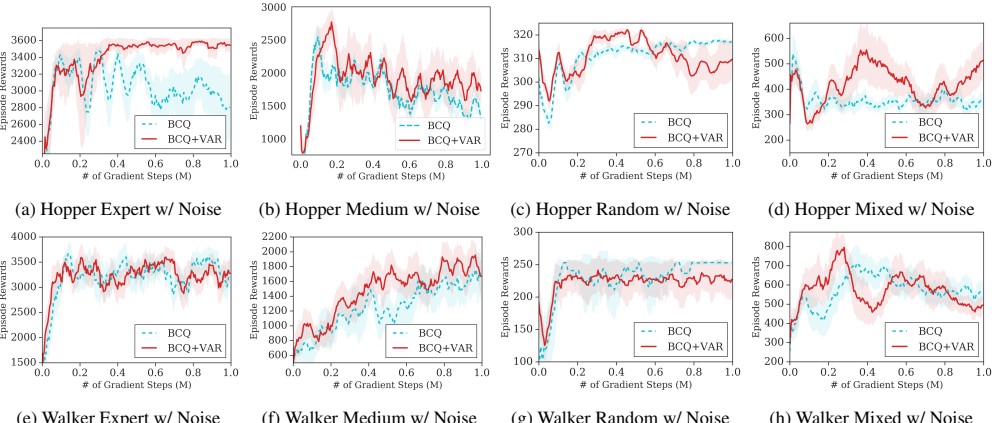

(a) Hopper Expert w/ Noise  (b) Hopper Medium w/ Noise  (c) Hopper Random w/ Noise  (d) Hopper Mixed w/ Noise

(e) Walker Expert w/ Noise  (f) Walker Medium w/ Noise  (g) Walker Random w/ Noise  (h) Walker Mixed w/ Noise

Figure 3: Evaluation of the proposed approach and the baseline BCQ on a suite of three OpenAI Gym environments. We consider the setting of rewards that are corrupted by a Gaussian noise. Results for the uncorrupted version are in Fig. 1. Experiment results are averaged over 5 random seeds

### E.3 Experimental Results on Safety Benchmark Tasks

**Safety Benchmarks for Variance as Risk :** We additionally consider safety benchmarks for control tasks, to analyse the significance of variance regularizer as a risk constraint in offline policy optimization algorithms. Our results are summarized in table 3.

### E.4 Discussions on Offline Off-Policy Optimization with State-Action Distribution Ratios

In this section, we include several alternatives by which we can compute the stationary state-action distribution ratio, borrowing from recent works (Uehara & Jiang, 2019; Nachum et al., 2019a).

**Off-Policy Optimization with Minimax Weight Learning (MWL) :** We discuss other possible ways of optimizing the batch off-policy optimization objective while also estimating the state-action density ratio. Following from (Uehara & Jiang, 2019) we further modify the off-policy optimization part of the objective $J(\theta)$ in $\mathcal{L}(\theta, \lambda)$ as a min-max objective, consisting of weight learning $\omega_{\pi/\mathcal{D}}$

Table 3: Results on the Safety-Gym environments Ray et al.. We report the mean and S.D. of episodic returns and costs over five random seeds and 1 million timesteps. The goal of the agent is to maximize the episodic return, while minimizing the cost incurred.

|  | PointGoal1 | | PointGoal2 | |
|---|---|---|---|---|
|  | Reward | Cost | Reward | Cost |
| BCQ | $43.1 \pm 0.3$ | $137.0 \pm 3.6$ | $32.7 \pm 0.7$ | $468.2 \pm 9.1$ |
| BCQ+OVR | $\mathbf{44.2 \pm 0.3}$ | $\mathbf{127.1 \pm 4.0}$ | $\mathbf{33.2 \pm 0.7}$ | $\mathbf{453.9 \pm 7.3}$ |

|  | PointButton1 | | PointButton2 | |
|---|---|---|---|---|
|  | Reward | Cost | Reward | Cost |
| BCQ | $\mathbf{30.9 \pm 2.2}$ | $330.8 \pm 8.3$ | $18.1 \pm 1.1$ | $321.6 \pm 4.1$ |
| BCQ+OVR | $30.7 \pm 2.3$ | $\mathbf{321.5 \pm 6.8}$ | $\mathbf{19.6 \pm 1.0}$ | $\mathbf{305.7 \pm 6.1}$ |

and optimizing the resulting objective $J(\theta, \omega)$. We further propose an overall policy optimization objective, where a single objective can be used for estimating the distribution ratio, evaluating the critic and optimizing the resulting objective. We can write the off-policy optimization objective with its equivalent starting state formulation, such that we have :

$$\mathbb{E}_{d_{\mathcal{D}}(s,a)}\Big[\omega_{\pi_\theta/\mathcal{D}}(s,a) \cdot r(s,a)\Big] = (1-\gamma)\mathbb{E}_{s_0 \sim \beta_0(s), a_0 \sim \pi(\cdot|s_0)}\Big[Q^\pi(s_0, a_0)\Big] \quad (66)$$

Furthermore, following Bellman equation, we expect to have $\mathbb{E}[r(s,a)] = \mathbb{E}[Q^\pi(s,a) - \gamma Q^\pi(s',a')]$

$$\mathbb{E}_{d_{\mathcal{D}}(s,a)}\Big[\omega_{\pi_\theta/\mathcal{D}}(s,a) \cdot \{Q^\pi(s,a) - \gamma Q^\pi(s',a')\}\Big] = (1-\gamma)\mathbb{E}_{s_0 \sim \beta_0(s), a_0 \sim \pi(\cdot|s_0)}\Big[Q^\pi(s_0, a_0)\Big] \quad (67)$$

We can therefore write the overall objective as :

$$J(\omega, \pi_\theta, Q) = \mathbb{E}_{d_{\mathcal{D}}(s,a)}\Big[\omega_{\pi_\theta/\mathcal{D}}(s,a) \cdot \{Q^\pi(s,a) - \gamma Q^\pi(s',a')\}\Big]$$
$$- (1-\gamma)\mathbb{E}_{s_0 \sim \beta_0(s), a_0 \sim \pi(\cdot|s_0)}\Big[Q^\pi(s_0, a_0)\Big] \quad (68)$$

This is similar to the MWL objective in (Uehara & Jiang, 2019) except we instead consider the bias reduced estimator, such that accurate estimates of $Q$ or $\omega$ will lead to reduced bias of the value function estimation. Furthermore, note that in the first part of the objective $J(\pi_\theta, \omega, Q)^2$, we can further use entropy regularization for smoothing the objective, since instead of $Q^\pi(s',a')$ in the target, we can replace it with a log-sum-exp and considering the conjugate of the entropy regularization term, similar to SBEED (Dai et al., 2018). This would therefore give the first part of the objective as an overall min-max optimization problem :

$$J(\omega, \pi_\theta) = \mathbb{E}_{d_\mu(s,a)}\Big[\omega_{\pi_\theta/\mathcal{D}}(s,a) \cdot \{r(s,a) + \gamma Q^\pi(s',a') + \tau \log \pi(a \mid s) - Q^\pi(s,a)\}\Big]$$
$$+ (1-\gamma)\mathbb{E}_{s_0 \sim \beta_0(s), a_0 \sim \pi(\cdot|s_0)}\Big[Q^\pi(s_0, a_0)\Big] \quad (69)$$

such that from our overall constrained optimization objective for maximizing $\theta$, we have turned it into a min-max objective, for estimating the density ratios, estimating the value function and maximizing the policies

$$\omega_{\pi/\mathcal{D}}^*, Q^*, \pi^* = \underset{\omega,Q}{\text{argmin}} \ \underset{\pi}{\text{argmax}} \ J(\pi_\theta, \omega, Q)^2 \quad (70)$$

where the fixed point solution for the density ratio can be solved by minimizing the objective :

$$\omega_{\pi/\mathcal{D}}^* = \underset{\omega}{\text{argmin}} \ \mathcal{L}(\omega_{\pi/\mathcal{D}}, Q)^2 = \mathbb{E}_{d_\mu(s,a)}\Big[\{\gamma\omega(s,a) \cdot Q^\pi(s',a') - \omega(s,a)Q^\pi(s,a)\}+$$
$$(1-\gamma)\mathbb{E}_{\beta(s,a)}Q^\pi(s_0,a_0)\Big] \quad (71)$$

**DualDICE :** In contrast to MWL (Uehara & Jiang, 2019), DualDICE (Nachum et al., 2019a) introduces dual variables through the change of variables trick, and minimizes the Bellman residual of the dual variables $\nu(s,a)$ to estimate the ratio, such that :

$$\nu^*(s,a) - \mathcal{B}^\pi \nu^*(s,a) = \omega_{\pi/\mathcal{D}}(s,a) \quad (72)$$

the solution to which can be achieved by optimizing the following objective

$$\min_\nu \mathcal{L}(\nu) = \frac{1}{2}\mathbb{E}_{d_{\mathcal{D}}}\Big[(\nu - \mathcal{B}^\pi\nu)(s,a)^2\Big] - (1-\gamma)\mathbb{E}_{s_0,a_0 \sim \beta(s,a)}\Big[\nu(s_0,a_0)\Big] \quad (73)$$

**Minimizing Divergence for Density Ratio Estimation :** The distribution ratio can be estimated using an objective similar to GANs (Goodfellow et al., 2014; Ho & Ermon, 2016), as also similarly

proposed in (Kostrikov et al., 2019).

$$\max_h \mathcal{G}(h) = \mathbb{E}_{(s,a)\sim d_{\mathcal{D}}}\Big[\log h(s,a)\Big] + \mathbb{E}_{(s,a)\sim d_{\pi}}\Big[\log(1 - h(s,a))\Big] \tag{74}$$

where $h$ is the discriminator class, discriminating between samples from $d_{\mathcal{D}}$ and $d_{\pi}$. The optimal discriminator satisfies :

$$\log h^*(s,a) - \log(1 - h^*(s,a)) = \log \frac{d_{\mathcal{D}}(s,a)}{d_{\pi}(s,a)} \tag{75}$$

The optimal solution of the discriminator is therefore equivalent to minimizing the divergence between $d_{\pi}$ and $d_{\mathcal{D}}$, since the KL divergence is given by :

$$-D_{\mathrm{KL}}(d_{\pi}||d_{\mathcal{D}}) = \mathbb{E}_{(s,a)\sim d_{\pi}}\Big[\log \frac{d_{\mathcal{D}}(s,a)}{d_{\pi}(s,a)}\Big] \tag{76}$$

Additionally, using the Donsker-Varadhan representation, we can further write the KL divergence term as :

$$-D_{\mathrm{KL}}(d_{\pi}||d_{\mathcal{D}}) = \min_x \log \mathbb{E}_{(s,a)\sim d_{\mathcal{D}}}\Big[\exp x(s,a)\Big] - \mathbb{E}_{(s,a)\sim d_{\pi}}\Big[x(s,a)\Big] \tag{77}$$

such that now, instead of the discriminator class $h$, we learn the function class $x$, the optimal solution to which is equivalent to the distribution ratio plus a constant

$$x^*(s,a) = \log \frac{d_{\pi}(s,a)}{d_{\mathcal{D}}(s,a)} \tag{78}$$

However, note that both the GANs like objective in equation 74 or the DV representation of the KL divergence in equation 77 requires access to samples from both $d_{\pi}$ and $d_{\mathcal{D}}$. In our problem setting however, we only have access to batch samples $d_{\mathcal{D}}$. To change the dependency on having access to both the samples, we can use the change of variables trick, such that : $x(s,a) = \nu(s,a) - \mathcal{B}^{\pi}\nu(s,a)$, to write the DV representation of the KL divergence as :

$$-D_{\mathrm{KL}}(d_{\pi}||d_{\mathcal{D}}) = \min_{\nu} \log \mathbb{E}_{(s,a)\sim d_{\mathcal{D}}}\Big[\exp \nu(s,a) - \mathcal{B}^{\pi}\nu(s,a)\Big] - \mathbb{E}_{(s,a)\sim d_{\pi}}\Big[\nu(s,a) - \mathcal{B}^{\pi}\nu(s,a)\Big] \tag{79}$$

where the second expectation can be written as an expectation over initial states, following from DualDICE, such that we have

$$-D_{\mathrm{KL}}(d_{\pi}||d_{\mathcal{D}}) = \min_{\nu} \quad \log \mathbb{E}_{(s,a)\sim d_{\mathcal{D}}}\Big[\exp \nu(s,a) - \mathcal{B}^{\pi}\nu(s,a)\Big] - (1-\gamma)\mathbb{E}_{(s,a)\sim\beta_0(s,a)}\Big[\nu(s_0,a_0)\Big] \tag{80}$$

By minimizing the above objective w.r.t $\nu$, which requires only samples from the fixed batch data $d_{\mathcal{D}}$ and the starting state distribution. The solution to the optimal density ratio is therefore given by :

$$x^*(s,a) = \nu^*(s,a) - \mathcal{B}^{\pi}\nu^*(s,a) = \log \frac{d_{\pi}(s,a)}{d_{\mathcal{D}}(s,a)} = \log \omega^*(s,a) \tag{81}$$

**Empirical Likelihood Ratio :** We can follow Sinha et al. (2020) to compute the state-action likelihood ratio, where they use a binary a classifier to classify samples between an on-policy and off-policy distribution. The proposed classifier, $\phi$, is trained on the following objective, and takes as input the state-action tuples $(s,a)$ to return a probability score that the state-action distribution is from the target policy. The objective for $\phi$ can be formulated as

$$\mathcal{L}_{cls} = \max_{\phi} -\mathbb{E}_{s,a\sim\mathcal{D}}[\log(\phi(s,a))] + \mathbb{E}_{s\sim\mathcal{D}}[\log(\phi(s,\pi(s)))] \tag{82}$$

where $s,a \sim \mathcal{D}$ are samples from the behaviour policy, and $s,\pi(s)$ are samples from the target policy. The density ratio estimates for a given $s,a \sim \mathcal{D}$ are simply $\omega(s,a) = \sigma(\phi(s,a))$ like in Sinha et al. (2020). We then use these $\omega(s,a)$ for density ratio corrections for the target policy in equantion **??**.

