# OpenReview forum: "Offline Policy Optimization with Variance Regularization"
_ICLR.cc/2021/Conference — Reject_

### Official Review · AnonReviewer2 · 2020-10-27
**The contribution is limited, and there are several technically unsound places**

**Rating:** 3
**Confidence:** 5

**Review:**


Summary:
This work is based on a recent work of [Bisi et al., 2019], where a per-step reward formulation is presented with some outstanding unresolved problems, e.g., the double-sampling issue and the policy-dependent reward issue. This paper proposes to use the Fenchel duality to solve the double sampling problem and extend it to the behavior-agnostic off-policy setting by leveraging the density ratio estimation technique.

Major concerns:

1
The derivations in several major equations are WRONG. The objective in Eq 5 is NOT the same as the objective in Eq 6. \E_{s, a ~ d_D}[\omega(s, a)r(s, a)] in Eq 6 is NOT equal to \E_{s~D}[Q^\pi(s, \pi(s)] in Eq 5.

2. The motivation and empirical demonstration of the variance regularization are unclear.
First, the definition of the variance doesn’t make sense to me. The variance in mean-variance optimization is a long-established term, which refers to the var of the return (either one-step or cumulative). So it is not clear why V_P makes sense without further motivation or reference. Moreover, the variance term defined in Eq. 2 is very weird. It is neither the variance of the return nor the so-termed “variance of the marginalized IS” (since it involves the reward r). By definition, the variance V_D is the variance of d_\pi(s, a)/d_\mu(s, a)r(s, a). This expression involves both \pi, \mu, and r, and its randomness comes from the randomness of (s, a). It is unclear why minimizing this variance is useful.

Second, the empirical results are not convincing. As pointed out by Eq. 3, the overall objective is to achieve a trade-off E[Q] and V_D(\pi) through \lambda. The empirical results, however, do not show this trade-off. Then it becomes unclear where the empirical improvement comes from. I would like to see how changing \lambda influences V_D(\pi).

3 Several key deductions, which hold in on-policy cases, may NOT be true in this paper’s off-policy setting. For example, Eq . 8. lacks proof. In Bisi’s setting, this holds only for on-policy cases. I don’t think it still holds for off-policy settings. The author needs to prove it.
The MDP setting is unclear. The authors consider an infinite horizon MDP, so what does T mean? The proof of Lemma 1 also seems problematic. First, without a clear definition of T, there is no way to check the proof of Lemma1. Is T a random variable? Second, Eq 33 is wrong. Eq 33 is the same as Eq 24, but the definition of D^\pi is different, so how can they be the same?
Moreover, It is hard follow the inequality in Eq 34. It looks wrong to me. Thee reviewer strongly suggest that the author write it step by step to make it clear? And also, it would be great to show how the products of IS in Eq 34 reduce to the density ratio in Eq 35.

4. Theorem 2, which is the paper's major theoretical contribution,  is obvious and trivial. By definition, the first term of RHS of Eq 16 is exactly J(\pi). So what theorem 2 says is that J(\pi) \geq J(\pi) \sqrt{c * variance of sth}}. This is fairly obvious and does not bring in any insight.

5. The entire Appendix B.2 is wrong, where the 3rd equality (aka, line 2 of Eq. 39) does NOT necessarily hold. The term d_\pi(\theta)’s gradient is not computed at all, and therefore, any results afterward are not correct.

6. There are some missing references as well. Using Fenchel Duality to solve the double-sampling issue in mean-variance optimization using variance as regularization has been solved by previous literature, e.g., Xie et al., (2018) and Zhang et al., (2020). The author should acknowledge this. Also, I encourage the authors to compare with them. Especially I think the authors may want to compare with Zhang et al., (2020). Algorithm 1 is very similar to the offline MVPI in Zhang et al., (2020). There is only a slight difference in computing the augmented reward.

Xie, T., Liu, B., Xu, Y., Ghavamzadeh, M., Chow, Y., Lyu, D., & Yoon, D. (2018). A block coordinate ascent algorithm for mean-variance optimization. In Advances in Neural Information Processing Systems (pp. 1065-1075).
Zhang, S., Liu, B., & Whiteson, S. (2020). Mean-Variance Policy Iteration for Risk-Averse Reinforcement Learning. arXiv preprint arXiv:2004.10888.

---

> ### Author Response · Authors · 2020-11-24
> **Response to AnonReviewer2 : Clarity of the Objective**
>
> Thank you for your detailed comments. We would like to clarify some of your misinterpretations of this work, and hope that we can clarify these doubts, and have an engaging discussion about some of your stated comments about this work.
>
> Firstly we would like to clarify the doubts stating the objectives in equation 5 and 6 are WRONG. This is indeed not the case, as we describe here. Equation 5 considers the regularized max-return objective, with the Fenchel duality trick applied to the variance regularization term. Following this, by noting the relations between the primal and dual forms, one can write $(1 - \gamma) E_{s \sim \beta_0, a \sim \pi(\cdot | s)} [ Q(s,a) ] = E_{(s,a) \sim d_{\pi}(s,a)} [ r(s,a) ]$, where $\beta_0$ is the initial state distribution and the R.H.S is the max-return objective under the occupancy measure. This equivalence is mentioned in a lot of related works using duality in RL (e.g Uehara et al., 2020). Following this, we can further apply the state-action distribution ratio correction $\omega$ to the objective under occupancy measures to denote the first term of eqn 6. To clarify the equivalence between primal and dual forms of the objective, we would encourage to please see (Nachum and Dai et al., 2020). By using this equivalence, and by using the dual form of the variance expression with distribution ratio corrections, we get the equivalence between eqn 5 and 6, which leads to overall augmented reward term in eqn 7. We would kindly encourage you to see the details of these related works, which will further justify why our objective in NOT WRONG.

---

> ### Author Response · Authors · 2020-11-24
> **Response to AnonReviewer2 : Discussion on related work**
>
> Thanks to AnonReviewer2 for pointing out that our work is related to (Zhang et al., 2020). We would like to emphasize that this work has been done parallel to ours, but was released in arxiv (without being published) a lot earlier. However, both these works were done at the same time, and we agree that the Fenchel duality trick in dealing with variance is similar in both works. However, we address the variance in different contexts, where our work is primarily in the off-policy offline RL setting, whereas Zhang et al proposes the approach for variance constrained actor-critic algorithms in the online setting. As such, the scope of experiments in Zhang et al is higher compared to ours, since their approach can be built on top of the many existing deep RL algorithms (e.g TRPO, SAC etc) while ours is more specifically focussed in the offline RL setting. Note however that both these works have the same issue, ie, avoiding the gradient of the state-distribution when optimizing w.r.t the dual form of the objective, as pointed out by AnonReviewer2. This is indeed a valid assumption/justification, in both this work and Zhang et al., we consider the occupancy measure to be insenstiive to the parameterized policy. Indeed, finding the gradient w.r.t the occupancy measure is difficult in practice. Similar assumption is also made in TRPO (Schulman et al., 2015) and further pointed out by (Ahmed et al., 2020 : PPO-DICE, see equation 2 and 3), following from (Kakade and Langford, 2002) which optimizes a surrogate objective based on the current policy and the state occupancy measure under the current policy. While we agree that this might be an assumption in this work too, we emphasize that this is a valid justification in most RL works considering policy improvement w.r.t the dual form of returns that avoids gradients of occupancy measures.
>
> Zhang, S., Liu, B., & Whiteson, S. (2020). Mean-Variance Policy Iteration for Risk-Averse Reinforcement Learning. arXiv preprint arXiv:2004.10888.

---

> > ### Author Response · Authors · 2020-11-24
> > **Clarity of derivation in Appendix and Further Discussion on Related Work**
> >
> > ```"The entire Appendix B.2 is wrong, where the 3rd equality (aka, line 2 of Eq. 39) does NOT necessarily hold. The term d_\pi(\theta)’s gradient is not computed at all, and therefore, any results afterward are not correct"
> >
> > We hope the above comment helps clarify this confusion. The gradient of the term $d_{\pi_{\theta}} is difficult to compute, and as also assumed in lot of other related works, we ignore the gradient w.r.t occupancy measure directly. The same assumption/justification is also used when deriving performance improvement bounds, e.g in TRPO or more recent PPO-DICE (Ahmed et al, 2020) and additionally more significantly related to our work by (Zhang et al., 2020). We believe finding the gradient w.r.t occupancy measure is itself an open and significant problem, which is beyond the scope of our work too. The inability to find this gradient term is also quite the reason why direct policy improvement based on the LP or extended LP formulation in RL is difficult, and most recent works along those lines simply considered policy evaluation rather then optimization. In our work, we consider policy optimization in the offline case, which further requires such assumptions. For clarity, we will include discussions on this in the updated version of the draft.
> >
> > "There are some missing references as well. Using Fenchel Duality to solve the double-sampling issue in mean-variance optimization using variance as regularization has been solved by previous literature, e.g., Xie et al., (2018) and Zhang et al., (2020). The author should acknowledge this. Also, I encourage the authors to compare with them. Especially I think the authors may want to compare with Zhang et al., (2020). Algorithm 1 is very similar to the offline MVPI in Zhang et al., (2020). There is only a slight difference in computing the augmented reward."
> >
> > We hope the above comment clarifies why we do not directly compare to Zhang et al., 2020 in our work too. The problem setup and domain in both these works are quite different, making empirical comparisons unfair. Furthermore, we would like to emphasize that both these works were done parallely at the same time, with the major difference that Zhang et al., 2020 appeared on arxiv earlier.

---

> ### Author Response · Authors · 2020-11-24
> **Response to AnonReviewer2 : Significance of Variance Regularizer**
>
> We would like to clarify the significance of the variance regularizer term we considered in this work. We emphasize that we denote V_P as the variance of the returns in the primal form (ie, the variance induced due the random variable r(s,a). In the off-policy case, with importance sampling corrections (unbiased estimator) this is still denoted as the variance V_P. In contrast, we denote the variance V_D in the dual sense, ie, when considering the objective w.r.t normalized state action occupancy measure. Again, in this case the variance is still due to the random variables r(s,a); however, in the off-policy case with distribution ratio corrections where this distribution ratio needs to be estimated, ie, $\omega(s,a)$, this will lead to additional variance in the dual form. For more details on variance term, please see Prashanth et al for example.
>
> The key idea of our work is to identify that the double sampling issue when considering the variance term can be avoided if we consider the duality view in RL; ie, if we consider the variance V_D term, it allows the use of Fenchel duality trick (similar to SBEED), which further leads to an augmented reward objective with the variance regularizer. Please also see the response to the other reviewers as to why this variance term is important/significant to consider in the offline RL case (mainly due to the variance induced by the distribution mismatch between the fixed dataset and the target policy). We hope this will clarify some of the doubts and help understand the significance of our work - which as we will additionally discuss - the aim is not to achieve superior empirical performance but rather to establish a long standing problem of how to deal with variance regularizers/constraints in policy optimization (e.g actor-critic with variance constraints).

---

### Official Review · AnonReviewer1 · 2020-10-28

**Rating:** 4
**Confidence:** 4

**Review:**

**Summary:**

This paper proposes a novel algorithm for offline policy optimization. The main idea is to prevent overestimation bias by regularizing against the variance of the importance weighted value estimate. There are two key modifications: (1) using an importance weight from the stationary distribution and (2) using Fenchel duality to introduce a min-max problem to avoid double sampling when estimating the gradient of the variance regularization term. The theory section motivates the use of variance regularization and the experiments show improvements over BCQ when adding the proposed variance regularization algorithm.

--------------------------------------------------------------------

**Strengths:**

1. The paper provides theoretical motivation for variance regularization. Theorem 2 demonstrates that using variance regularization can be seen as optimizing a lower bound on the true value of a policy.
2. The algorithm provides a novel way to implement variance regularization. By introducing the $ \nu $ variables, the paper proposes a way to get around double sampling issues in estimating the gradient of the regularizer.

--------------------------------------------------------------------

**Weaknesses:**

1. There are several technical steps that I am not convinced by. I am not sure if these are mistakes or just statements that require more explanation:
   1. In the algorithm, $ \nu $ is defined by $ \tilde r $ and $ \tilde r $ is defined by $ \eta $ in equation (7). This definition seems somewhat circular and the paper never clearly explains why this should work.
   2. The continual updating of $ \tilde r $ seems to make the problem nonstationary for the policy improvement subroutine that depends on the current iterate of the distribution ratio $ \omega_\psi$. The argument made in a remark on page 4 does not seem to sufficiently resolve this issue since even though $ \nu $ has a closed form, that closed form depends on the distribution ratio which depends on the current policy.
   3. The variance term that shows up in Theorem 1 does not seem to be what is being estimated by $ V_D $ in the algorithm. In light of this, what is the point of Theorem 1 vis a vis the proposed algorithm?
   4. Theorem 2 shows that using the square root of the variance as a regularizer provides a lower bound, but the algorithm uses the variance. What is the explanation for this mismatch?
2. Notation is somewhat confusing and sloppy. For example, $J$ is defined in several different ways, first in equation (1) and then in equation (3). This makes the rest of the paper confusing since it is sometimes unclear which $J$ is being referred to. Another example is that $ V_D $ is defined as a function of $ \pi $ in equation (2) and then a function of $ \omega, \pi $ in equation (3). Equation (4) just refers to $ V $ when I think it means $ V_D$. Also in equation (4) the LHS has no $s,a$ variables but the first two terms on the RHS are a function of $ s,a $. Where do these $s,a$ come from? Is there an expectation missing? In equation (5) it is not clear which $ \omega $ is being referred to. Equation (6) overloads $J$ once more.  Equation (6) could also use a better explanation as to why this is the dual form of (5) rather than just asserting it. The definitions of $ \epsilon_\pi, L_\pi $ are not included in Lemma 2, but are instead in the following paragraph. In equation (12) $ d_\pi $ is denoted by $\beta_{\pi'}$ for no apparent reason. In equation (13), $f$ is supposedly a dual class of functions, is there a sup missing? In equation (15), ${\hat \omega_{\pi /D}}$ is never explicitly defined. This sort of sloppiness pervades the paper and makes it often difficult to understand.
3. The writing is unclear. One major issue is that lemmas are presented with no context or direction. It would be helpful to preface each section with an explanation of where the section is going before presenting technical lemmas. This is a problem for both Lemma 2 and Lemma 3 where it is unclear to the reader what the point of the lemma is until much later in the section. This is also a problem for Lemma 1 which has it's own subsection that does not seem to make any clear claim to connect to the thesis of the paper. The lemmas may be better suited in the appendix rather than the main text of the paper or just require more explanation.
4. Empirical results are not very convincing. The plots presented in the experimental section seem to show a slight but not large or consistent advantage for the proposed method. Perhaps more worrying, the only type of result reported are the final performance of the algorithms. There is no empirical indication as to how the algorithm is working or whether the variance regulation is indeed having the desired effect of reducing overestimation. Moreover, there is no indication that the Fenchel duality tricks are doing anything useful empirically.
5. Important ablations and baselines are missing. The authors do not include conservative Q learning (Kumar et al., 2020) as a baseline saying that it is too recent to compare. However, this paper came out in June and according to the ICLR reviewer guidelines we are only meant to consider work released in August or later as contemporaneous. So, CQL ought to be included as a baseline and I think generally outperforms the proposed methods. Additionally, while there is one ablation included in the appendix (adding importance weighting to BCQ), more ablations are needed. Specifically, since the main algorithmic innovation of the paper not just using variance regularization, but computing gradients using the Fenchel duality max-min procedure, there should be ablations showing whether this part of the algorithm is indeed necessary and useful.

--------------------------------------------------------------------

**Recommendation:**

I recommend rejection and gave the paper a score of 4. While there may be a good idea in there, I do not think the paper is ready to publish. A more clear and careful exposition of the algorithm as well as more rigorous theory and experiments are needed.

--------------------------------------------------------------------

**Additional feedback:**

Typos:

- The first sentence of the intro has a space before the period.
- The references on the top of page 2 seem to be incorrect. "A." is the first initial of the first author, not their last name.
- Throughout the paper there is usually a space preceding each colon. There should be no spaces before each colon.
- After equation two "a as" should be "as a "
- In the first sentence of section 3.4 "leads" should be "lead"
- The paper sometimes refers to the proposed variance regularization as a "framework" (e.g. in the first sentence of the conclusion). It is not a framework, it is an algorithm.

---

> ### Author Response · Authors · 2020-11-23
> **Response to AnonReviewer1 : Clarification**
>
> Thank you for your detailed review and we appreciate your comments about the significance of our work.
>
> We would like to clarify that the key contribution/motivation of our work is not to avoid the overestimation bias in offline RL, unlike addressed by lot of recent works. We rather propose an algorithmic approach to constrain the policy improvement step in offline RL with variance which we note is important because of the induced distribution shift in offline RL (due to the mismatch between fixed data distribution collected by the behaviour policy and the target policy). The key idea of our work is to show how to avoid double sampling issues when constraining offline policy optimization algorithms with variance of returns, and we address this in context of marginalized importance weighted corrections (ie, correcting for state-action distribution ratios).
>
> Augmented rewards and non-stationarity : As pointed in equation 7, the proposed regularizer and the use of the Fenchel duality trick leads to the augmented reward objective. We would like to emphasize that this is a desired result, since often with variance regularization, policy improvement is difficult and previous works have taken a primal-dual approach with zero duality gap. In contrast, we view the variance as a regularizer, and show that the duality tricks allows for eventually only augmenting the rewards with the dual variables, which is further solved in an optimization step (due to the closed form solution of the dual variables). While we appreciate your comment regarding non-stationarity, and you are correct in pointing this out - note that this is not as difficult as it seems, since the inner loop has a closed form solution of dual variables, and the distribution ratio is further solved in an inner optimization step. Therefore, the use of alternating updates avoids the non-stationarity issue, since when optimizing for returns, only the reward function with the given distribution ration and dual variables is used. This is therefore equivalent to any other works in RL which uses augmented rewards (e.g entropy regularization when the entropy is with the changing parameterized policy in policy gradient methods).
>
> Regarding the variance term and Theorem 1 : Note that the presented theoretical results are achieved due to the variational representation of divergences and the use of Pinsker’s inequality. Theorem 1 is presented to establish the significance of guaranteed policy improvement step, even under the presence of variance. Compared to that V_D is denoted as the variance of off-policy corrected returns, and V_P as the variance of returns (ie, in the dual and primal forms respectively). The presented policy improvement step applies for both on-policy or off-policy, through the use of variational representation - and we presented the results in section 4 in the generic sense since it applies broadly.
>
> Minor typos and Revisions : We hope to address your minor concerns regarding typos and unclear sentences in the updated version of our draft. Thank you for pointing out the ambiguity in notations - we will address these in the revised version. To be precise, we should denote eqn 1 as the usual max-return objective and all other follow-ups are the regularized max-return objective with the regularizer. Similarly, V_P is often denoted as the primal variance of returns term (variance of sum of discounted returns), while V_D denotes the variance in dual form (ie, w.r.t occupancy measures * rewards). We denote $\hat{\omega}$ to specify that it needs to be estimated from the inner loop optimization. Regarding the presentation of theoretical results, we appreciate your useful feedback and would try to be more precise in the updated version such that the main thesis of the paper is easy to follow. This will hopefully make it clear why the theoretical results are presented this way (for more details, please also see the response to AnonReviewer4 about theoretical results section 4). We hope that the novelty and the significance of our work, and the impact it has in the overall RL community can be emphasized more than the minor typos which can be addressed in the revised version of the draft. We believe our work is important in addressing a major problem dealing with variance of returns in off-policy RL.

---

> ### Author Response · Authors · 2020-11-23
> **Response to AnonReviewer1 : Comments on experimental results and baselines**
>
> We emphasize that the key contribution of the work is to avoid the double sampling issue in policy optimization when constraining or regularizing with the variance of off-policy corrected returns. Our aim was not to improve benchmark results compared to existing offline RL algorithms and as such we are not aiming to provide a new offline RL algorithm that avoids overestimation bias, unlike BCQ/CQL that can achieve significant empirical performance on standard tasks. In contrast, the key thesis of this work is to show that even under presence of explicit variance constraints, using the dual form with Fenchel duality trick helps avoid the double sampling issue and empirically can lead to a well performing algorithm on the standard offline benchmarks.
>
> Our proposed algorithmic approach can be applied on top of any existing offline RL algorithm. We chose BCQ as a baseline algorithm to build on top of, due to the open source implementation and the ease of use of this algorithm. In contrast, even though CQL came out recently, on repeated requests with the authors, the code of CQL was not released at the time of submission of this work. Similar variance regularizers can also be used on top of CQL to present the proposed benefits, but as we pointed out before - our aim is not to achieve SOTA results on offline RL tasks and we do not aim to compare to all algorithms to justify the importance of the regularizer. Our proposed approach is rather more motivated from a theoretical point of view, to emphasize the need for accounting for constraints, especially those of variance due to the induced distribution shift, which is important to address when scaling up offline algorithms in practical scenarios.
>
> Overall : We hope our detailed response will clarify some of your doubts about this work and we apologize for the confusion that this may have caused. We believe we have addressed your comments and this will help you to get better insights about our work and the significance it has in offline RL related works. We emphasize that our contribution in this work is to demonstrate the use of variance regularization in offline RL, which can often be difficult from an optimization perspective, and our work hopes to shed light in addressing some of these issues, while achieving policy improvement guarantees even under the presence of variance constraints. We hope our detailed response will help you in reflecting on the overall score of this work.

---

### Official Review · AnonReviewer4 · 2020-11-06
**The paper proposes an interesting idea, but it needs revision.**

**Rating:** 4
**Confidence:** 3

**Review:**

Disclaimer: this paper was assigned to me as an emergency review paper, so I might be missing something important.

### Evaluation
I recommend rejection. I think the paper presents an interesting idea to regularize policy updates by the variance of a return. Besides, it proves that the return-variance-regularized policy update leads to a monotonic policy improvement, which I think is novel. That being said, the paper seems to have several issues that decreased my evaluation. First, the clarity of the paper is really low: many typos, ambiguous notations, and confusing presentation of a new algorithm. Second, I could not understand some parts of the paper, especially a proof of the monotonic policy improvement theorem. Third, it is unclear why the proposed regularizer is suitable for offline RL.

### Paper Summary
Offline RL is (said to be) a key to enable RL applications to real world problems. The paper proposes a new algorithm, which regularizes policy updates by the variance of return, for offline RL. The paper proves a monotonic policy improvement when the return-variance-regularized policy update is used. Experiments show moderate performance gain by the regularization.

### Strong Points
1. Interesting idea to regularize policy updates by the variance of a return
2. Moderate performance gain, especially when offline dataset is obtained by a suboptimal policy.

### Weak Points
1. The paper is not well-written. It contains typos, unclear sentences, and ambiguous notations.
2. Some parts of the paper, like the proof of the monotonic policy improvement theorem, seem to contain mistakes. (I might be wrong, though.)
3. The performance gain seems to be moderate, despite a high complexity of the computation of the proposed regularizer.

### Comments to the Authors

If I am missing something, please feel free to let me know. As noted above, I could not spare much time to review the paper.

1. The paper is not well-written. Please revise it again. (I don't point out all ambiguous notations and typos, but later I point out some serious ones.) Besides, please revise the references section. It refers to arxiv versions of papers that were accepted to conferences.
2. In page 3, $\omega_{\pi / \mathcal{D}}$ suddenly appears. What does it mean? Maybe $\omega_{\pi / \mu}$?
3. What does $s \sim \mathcal{D}$ mean? Does it mean $s \sim d_\mu$? Since the dataset $\mathcal{D}$ contains states visited by $\mu$, simply drawing states from $\mathcal{D}$ will be different from $d_\mu$.
4. What is $d_{\mathcal{D}}$ in, for example, Equation 4?
5. The idea to use the Fenchel duality to avoid double-sampling problem seems to be not new (cf SBEED paper). While the paper mentions AlgaeDICE as an algorithm using a similar technique, it does not mention SBEED about the use of Fenchel duality. Why?
6. In the beginning of Section 3.4, the min-max problem is being solved by repeating the inner minimization and outer maximization. As far as I remember (I might be wrong, though!), this way of solving a min-max problem might not find the exact solution. Isn't it a problem?
7. In Equation 6, it seems that $Q^\pi (s, a)$ is rewritten as $E_{(s, a) \sim d_\mathcal{D}} [ \omega (s, a) r (s, a)]$. According to the notation of the paper, isn't $E_{(s, a) \sim d_\mathcal{D}} [ \omega (s, a) r (s, a)]$ be $(1-\gamma) E_{s_0 \sim \beta, a_t \sim \pi} [\sum_{t=0}^\infty \gamma^t r(s_t, a_t)] \neq Q^\pi(s, a)$?
8. Is Lemma 1 different from Lemma 1 of [Bisi et al, 2019](https://arxiv.org/pdf/1912.03193.pdf)? Also what is its meaning? Why is is useful to understand the variance regularizer?
9. How is the beginning of Section 4.2 related to Theorem 1? Theorem 1 seems to be derived based on a different inequality in its proof.
10. As far as I remember, the dual form of the total variation is $\sup_{f \in C_`1} E_{A \sim P} f(A) - E_{B \sim Q} f(B)$, where $C_1$ is the space of all continuous functions bounded by $1$. Therefore, we don't need $\phi$ and can explicitly state the space of $f$ in Equation 12. Am I wrong or missing something?
11. Why is there no sup over f in Theorem 1?
12. As for Equation 59, how do you get the first line? In addition, do you need $E_{s \sim d_\pi}$? In the second line, the first $E_{s \sim d_{\pi'}, a \sim \pi}$ is sampling an action from $\pi$. Isn't it $\pi'$? In the fourth line, $d_{\pi'}$ changed to $d_\pi$. How is it possible?
13. I don't fully understand what Section 4.3 does. In addition, what are random variables in Theorem 2?
14. I don't understand what the final objective to be optimized is. In Algorithm 1, $J(\theta, \phi, \psi, \nu)$ appears. Is it the same as Equation 6?

---

> ### Author Response · Authors · 2020-11-23
> **Reply to AnonReviewer4 : Detailed comments to improve clarity and discuss significance of our work**
>
> Thank you for your detailed comments and we appreciate your feedback. Here, we will try to address some of your concerns address your comments to help improve your understanding of our work.
>
> The key to this work is to propose a variance constrained policy optimization approach, which we believe is important to address especially for offline RL. This is mostly because often the data collecting behaviour policy may induce large distribution shift with the target policy under optimization, and as such, the variance of returns can often lead to deterioration of performance. We specifically address this in context of marginalized importance sampling (Liu et al., 2018) which has recently gained interest in off-policy evaluation (Nachum et al., 2019). However, previous works have only addressed this in off-policy evaluation setting, while we extend this in the optimization setting (ie, policy improvement under batch data) with the additional challenge of constraining the variance of marginalized importance corrected returns. Dealing with variance of returns has been difficult in RL in general, especially for policy improvement or actor-critic algorithms since it leads to the double sampling issue due to the variance term.
>
> Performance Gains : We would like to emphasize and note that the significance or primary aim of our work was not to significantly improve performance of offline RL algorithms. Lot of existing deep RL works try to address performance gains, which comes with drawbacks and practical instabilities too (Henderson et al., 2017) and our aim was not to beat all offline RL benchmarks. Instead, the primary contribution of our work was to present a simple, novel, yet effective idea with a cute little trick that can address algorithmic difficulties in dealing with variance of off-policy corrected returns in RL.
>
> We denote $w_{\pi/\mathcal{D}}$ to denote the marginalized importance weighting between the state-action distribution of the target policy and the fixed data distribution. In the draft, we initially denoted this as $w_{\pi/\mu}$ for the generic case when the behaviour off-policy  is known, while w_{\pi/\mathcal{D}} denotes for any known or unknown fixed data distribution. Additionally, $d_{\mathcal{D}}$ denotes the joint state-action distribution in the dataset, where data can be collected by the known or unknown behaviour policy $\mu$. You are correct in addressing the SBEED paper which uses the Fenchel duality to address the double sampling issue. Instead of specifically citing the SBEED paper, we in fact referred to the more generic paper (Nachum & Dai et al., 2020) which discusses all the related approaches using duality tricks in RL.
>
> We pointed the adaptation of Lemma 1 from Bisi et al., and included the proof for completeness in Appendix. The significance of presenting Lemma 1 is to emphasize that difference between variance of returns in the primal and dual forms. This is further established by equations 33-35 in the appendix, extending Lemma 1 to the off-policy case with known behaviour policy $\mu$. This is to emphasize that regularizing the policy improvement step with variance in the dual form leads to a lower bound and it comes with benefits from an optimization perspective (ie, avoiding the double sampling issue).
>
> We would like to emphasize that the first part of section 4.2 was to first mention the related approaches for policy improvement bounds based on the TV/KL as appeared in related works (ie, equations 10 and 11). Theorem 1 is presented in such a way to highlight that we can similarly derive performance improvement bounds with variance regularization, compared to related works which derives this with the divergence measures - this is where we mention the use of Pinsker’s inequality that is useful following the first presented part of section 4.2. Regarding the dual form of the variational representation, you are correct in pointing this out, and we only denote $\phi$ to be the convex conjugate of the functions f. Finally, section 4.3 follows from this to derive the lower bound performance improvement result which appears due to the use of variance regularizer and duality tricks. The first line of equation 59 comes due to the use of Pinsker’s inequality, which states the relation between the TV and KL divergences. The first line of eqn 59 therefore follows from eqn 58. Furthermore, the E in equation 59 are not typos, but are rather standard, since it derives from the use of the performance difference lemma (Kakade et al).
>
> Overall : We emphasize that our contribution in this work is to demonstrate the use of variance regularization in offline RL, which can often be difficult from an optimization perspective, and our work hopes to shed light in addressing some of these issues, while achieving policy improvement guarantees even under the presence of variance constraints. We hope our detailed response will help you in reflecting on the overall score of this work.

---

### Decision · Program_Chairs · 2021-01-07
**Final Decision**

**Decision:**

Reject

**Comment:**

The reviewers are unanimous that the submission does not clear the bar for ICLR.